# Genetic control of meristem arrest and life span in *Arabidopsis* by a *FRUITFULL-APETALA2* pathway

Vicente Balanzà[1], Irene Martínez-Fernández[1], Shusei Sato[2,5], Martin F. Yanofsky[2], Kerstin Kaufmann[3], Gerco C. Angenent[3,4], Marian Bemer[3] & Cristina Ferrándiz[1]

Monocarpic plants have a single reproductive cycle in their lives, where life span is determined by the coordinated arrest of all meristems, or global proliferative arrest (GPA). The molecular bases for GPA and the signaling mechanisms involved are poorly understood, other than systemic cues from developing seeds of unknown nature. Here we uncover a genetic pathway regulating GPA in *Arabidopsis* that responds to age-dependent factors and acts in parallel to seed-derived signals. We show that *FRUITFULL* (*FUL*), a MADS-box gene involved in flowering and fruit development, has a key role in promoting meristem arrest, as GPA is delayed and fruit production is increased in *ful* mutants. *FUL* directly and negatively regulates *APETALA2* expression in the shoot apical meristem and maintains the temporal expression of *WUSCHEL* which is an essential factor for meristem maintenance.

[1] Instituto de Biología Molecular y Celular de Plantas, Consejo Superior de Investigaciones Científicas—Universidad Politécnica de Valencia, 46022 Valencia, Spain. [2] Division of Biological Sciences, University of California San Diego, La Jolla, CA 92093, USA. [3] Laboratory of Molecular Biology, Wageningen University & Research, Wageningen, 6708 PB, The Netherlands. [4] Bioscience, Wageningen University & Research, Wageningen, 6708 PB, The Netherlands. [5] Present address: Graduate School of Life Sciences, Tohoku University, Sendai 980-8577, Japan. Correspondence and requests for materials should be addressed to C.Fán. (email: cferrandiz@ibmcp.upv.es)

Life-history in plants or animals can be classified in two fundamentally different strategies: iteroparity, when the individual reproduces multiple times in life; and semelparity, when offspring is the result of a single reproductive cycle, and the individual dies shortly after. Examples of semelparous plants, also known as monocarpic, are found in all angiosperm groups, including the model species *Arabidopsis thaliana* and crops of huge economic importance such as grain legumes or cereals. In many monocarpic plants, after the production of a certain number of fruits, all meristem activity arrests coordinately (undergoes global proliferative arrest, or GPA) preceding completion of fruit filling and subsequent plant death, a strategy usually recognized as a way to optimize allocation of resources to the production of seeds[1–3]. It is well established that fruit and seed production are major factors controlling meristem arrest in different species, including *Arabidopsis*[1–3]. Moreover, in addition to the signaling from developing fruits, it has also been proposed that GPA timing is genetically controlled by an age-dependent pathway, as indicated by the different timing of meristem arrest and total fruit production in different ecotypes of *Arabidopsis*[1]. However, little is known about the molecular and genetic mechanisms controlling GPA timing.

In this study, we have identified two mutants that show delayed GPA and act in parallel to the seed-dependent cues affecting proliferative arrest. We show that *FUL* is expressed in the inflorescence meristem where it directly and negatively regulates the accumulation of *AP2* and *AP2*-like genes, which ultimately have an effect in the maintenance of *WUS* expression in the meristem and, hence, in its proliferative capacity. Our work sheds some light on the age-dependent control of GPA, which so far has been poorly understood.

## Results

### *ful* mutants have delayed GPA.

We observed that *Arabidopsis fruitfull* (*ful*)[4] mutants produced more fruits than wild-type before senescing regardless of the genetic background, as it could be observed for two independent alleles, *ful-1* (L*er*) and *ful-2* (Col), suggesting that *FUL* could be involved in GPA regulation (Fig. 1a, b). The increase in fruit number was due to an extended period of flower production, which in wild-type Columbia plants proceeded at decreasing rates for approx. 4 weeks, while *ful-2* mutants continued to produce flowers for 7–8 weeks before GPA took place (Fig. 1c, Supplementary Figure 1). This is different from previously reported mutants showing increased flower production such as those affecting cytokinin degradation, where meristems are larger and produce more flowers, but arrest at the same time as wild-type[5].

It has been described that GPA is controlled by seed production. In *Arabidopsis*, when the number of seeds produced per fruit is reduced to less than 30–40% of wild-type, GPA is delayed and meristem growth eventually terminates by the differentiation of a terminal flower mainly of carpeloid nature[1]. In *ful* mutants, seed set is around 70% of wild-type (Fig. 1b), but this does not appear to be the cause of the delayed GPA, because unrelated mutants with a similar reduced seed set, like *hecate3* (*hec3*; seed set of around 63%)[6] or *feronia (fer)* heterozygotes (female gametophyte mutant, seed set around 50%)[7], produce a similar number of flowers and terminate like wild-type (Fig. 1b). In addition, we compared flower production in wild-type and *ful-2* where seed production was fully avoided by manual removal of all flowers as they were produced by the SAM. Seedless wild-type plants produced an average of 70 floral nodes before differentiating into a terminal flower, while *ful-2* mutants produced an average of 92 floral nodes and arrested without producing a terminal flower (Fig. 1d, e). Together these experiments suggest that the increased number of flowers in *ful* mutants is not related to fertility defects.

### *WUS* repression in the SAM correlates with GPA.

WUSCHEL (WUS) is a homeodomain transcription factor expressed in the organizing center of the shoot apical meristem (SAM), where it is essential for the maintenance of the stem-cell pool[8, 9]. To monitor *WUS* expression in inflorescence meristems we used whole mount RNA in situ hybridization and a WUS::GUS reporter line[10] at different time points in either L*er* or Col backgrounds. Although both can only be described as semi-quantitative techniques to provide spatial and temporal resolution, they were preferable to direct quantification of *WUS* expression levels by

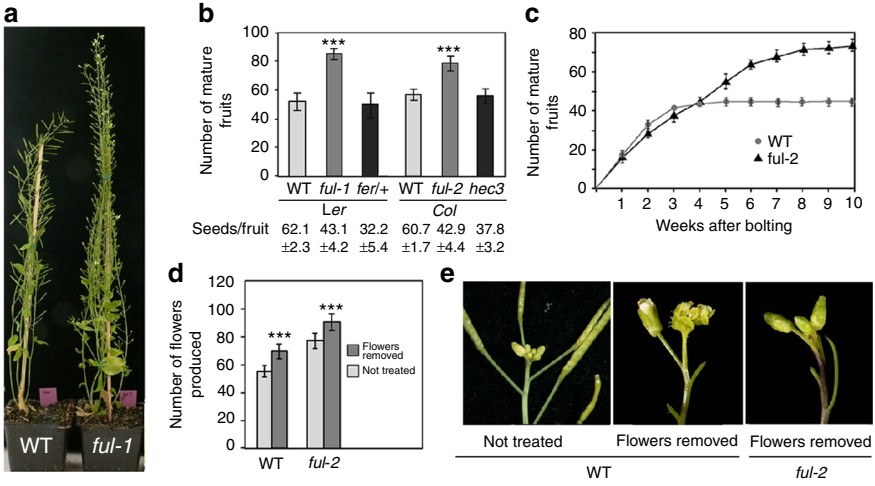

**Fig. 1** FUL regulates GPA timing. **a** *ful* mutants produce more fruits than WT. **b** The increase in the number of fruits produced by the main inflorescence in *ful* mutants does not depend on the genetic background or on the reduced seed production of *ful* fruits, since other mutants with similar reduced fertility (like *hec3* or *fer/+*) undergo GPA-like WT. Asterisks (***) indicate a significant difference (*P* < 0.001) from WT according to Student's *t* test. **c** *ful* mutants produce more fruits because GPA is delayed and the period of fruit production is extended. **d** Induced sterility by continuous manual pruning of developing flowers causes increased flower production in WT and *ful* mutants. Asterisks (***) indicate a significant difference (*P* < 0.001) from the corresponding untreated plants according to Student's *t* test. **e** While sterility in WT leads to the formation of a terminal flower, in *ful* mutants sterility only leads to a further delayed GPA. *n* = 10 for all genotypes in **b**, **c**, **d**

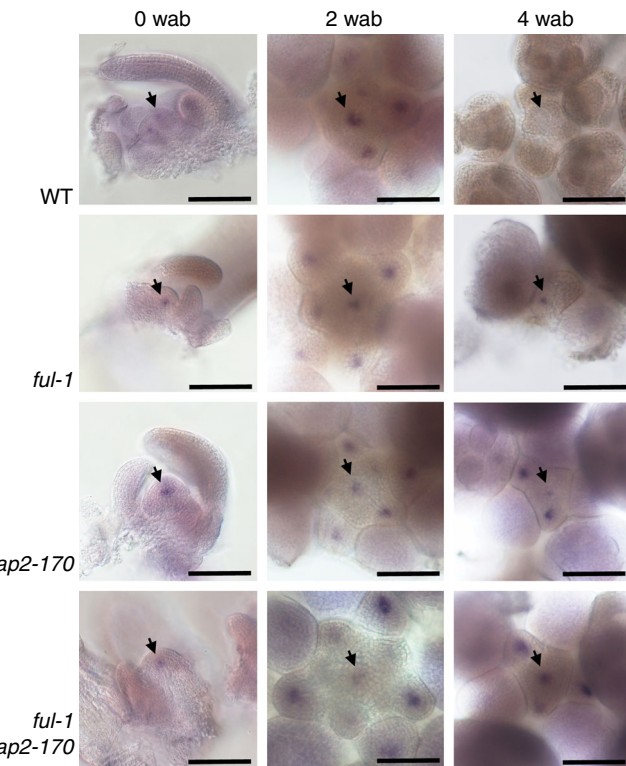

**Fig. 2** *WUS* expression is temporally extended in the SAM of *ful* and *ap2-170* mutants. Whole mount in situ hybridization of *WUS* mRNA in SAMs from different genetic backgrounds shortly after floral transition (0 weeks after bolting- 0 wab), 2 weeks after bolting (2 wab) and 4 weeks after bolting (4 wab), a time point when wild-type plants have undergone GPA and stop producing new floral meristems, while mutants are still proliferative. Arrows indicate the SAM. Bars = 100 μm

qRT-PCR for our purposes as that would have to be performed in dissected SAMs in order to avoid inclusion of floral tissue and because SAMs in older stages are extremely reduced in size (Supplementary Figure 1). *WUS* expression was detected at the center of the wild-type SAM and decreased with meristem age, being no longer detectable in arrested meristems (Fig. 2, Supplementary Figure 2a, b), while still present in developing ovules of non-arrested floral buds in these plants (Supplementary Figure 2a). In *ful* mutants, *WUS* expression was similar to wild-type at earlier time points, but was still present in *ful* inflorescences at 4 or 5 weeks after bolting, and even later time points, long after wild-type plants had undergone GPA (Fig. 2, Supplementary Figure 2a, b), thus showing the correlation of *WUS* decline with meristem arrest and suggesting that FUL controls GPA through the temporal repression of *WUS* in the SAM. In addition to *WUS*, we also monitored the expression of the meristem genes *CLAVATA3* (*CLV3*) or *SHOOT MERISTEMLESS* (*STM*) using CLV3::GUS and STM::GUS reporter lines[11, 12]. As described for *WUS*, *CLV3* and *STM* showed similar expression in young wild-type or *ful* inflorescence meristems, decreasing with age in both backgrounds, but more rapidly in wild-type SAMs than in *ful* mutants (Supplementary Figure 2b). Interestingly, both *CLV3* and *STM* could still be detected at time points when WUS::GUS was no longer expressed, further suggesting that *WUS* repression marked GPA timing.

**FUL regulates GPA by fine-tuning AP2 activity in the SAM**. To further elucidate the mechanisms by which FUL can control GPA, we performed an EMS enhancer-suppressor mutagenesis

screen on *ful-1*[13]. A mutant line, 170.2, showed dramatically extended life span of the SAM, with plants producing more than twice as many fruits than wild-type, that usually died of external causes before GPA could take place. In addition, the L170.2 mutants showed a strong loss of floral meristem determinacy (Fig. 3a, b). Through a map-based cloning approach, we found that line 170.2 carried a G to A nucleotide substitution in the last exon of the *APETALA2* (*AP2*) gene which affected the microRNA binding site of miR172[14] and caused a G398E transition in the AP2 protein, so we renamed this allele *ap2-170* (Supplementary Figure 3). When segregated from the *ful-1* mutant background, the *ap2-170* mutation caused a significantly delayed GPA compared to wild-type L*er* (Fig. 3a) and fertile fruits that were 20% shorter than wild-type, but no other defects related to the described roles of AP2 in floral organ development or floral determinacy (Fig. 3b)[15–17]. The progeny of *ap2-170* crossed to *ap2-2*, a loss-of-function *AP2* allele showing homeotic conversion of the floral organs, produced flowers similar to wild-type, with no homeotic defects, suggesting that the AP2[170] protein was largely functional (Fig. 3b). Interestingly, AP2 has been identified as a positive regulator of *WUS*[17–19], likely by an indirect mechanism, and therefore, the extended *WUS* expression observed in the SAM of *ful* mutants could be related to these changes in AP2 activity. In fact, *ap2-170* mutants also showed persistent *WUS* expression in the SAMs 4 weeks after bolting, similar to what was observed in *ful* mutants (Fig. 2), expression that was further extended in the *ful-1 ap2-170* double mutant (Fig. 2).

Additional evidence for the role of AP2 in preventing GPA and maintaining meristem activity was obtained using a dexamethasone-inducible AP2[170] line (pOpON:AP2[170], Supplementary Figure 4). AP2[170] induction in senescent wild-type plants that already had undergone GPA restored the activity of the SAM, which produced several new flowers and fruits before arresting again (Fig. 3c).

Since the strong floral indeterminacy phenotype was only observed in the *ful* background and the delay in GPA was dramatically enhanced in the double *ap2-170 ful* mutant, it appeared that *ful* mutations could affect the activity of the *ap2-170* allele, and furthermore, it revealed that *FUL* had a role in stem cell regulation both in the SAM and the floral meristem. We generated transgenic lines harboring either 35S::AP2[170] or 35S::AP2 constructs in both wild-type and *ful* mutant backgrounds. AP2[170] overexpression induced a strong loss of floral determinacy similarly in wild-type and in *ful* mutants, while 35S::AP2 did not cause visible defects in any of these backgrounds (Fig. 3d), suggesting that AP2 acts downstream of FUL, and that the mutation in the miR172 target site found in *ap2-170* is important for *AP2* regulation in meristems. To study whether FUL is regulating *AP2* expression, we quantified *AP2* mRNA levels in dissected inflorescence apices of wild-type and *ful* mutants 3 weeks after bolting, following removal of all visible floral buds under the scope. *AP2* expression levels were significantly elevated in *ful* SAMs, suggesting that FUL is an *AP2* repressor (Fig. 3e).

To investigate whether FUL could directly bind to *AP2*, we searched a chromatin immunoprecipitation (ChIP) experiment with plants that expressed pFUL::FUL:GFP in the *ap1 cal* mutant background. The inflorescences of these lines are composed of proliferative inflorescence meristems (GEO accession number GSE108455). In this ChIP-seq data, we identified binding sites of FUL upstream of *AP2* as well as upstream of the other miR172-regulated *AP2* homologs *SCHNARCHZAPFEN* (*SNZ*), *TARGET OF EARLY ACTIVATION TAGGED* (*EAT*) *1* (*TOE1*), and *TOE3* (Supplementary Figure 5)[14, 20]. No enrichment could be observed at the *WUS* locus (Supplementary Figure 5), indicating that FUL does not regulate *WUS* directly. To confirm these data, we

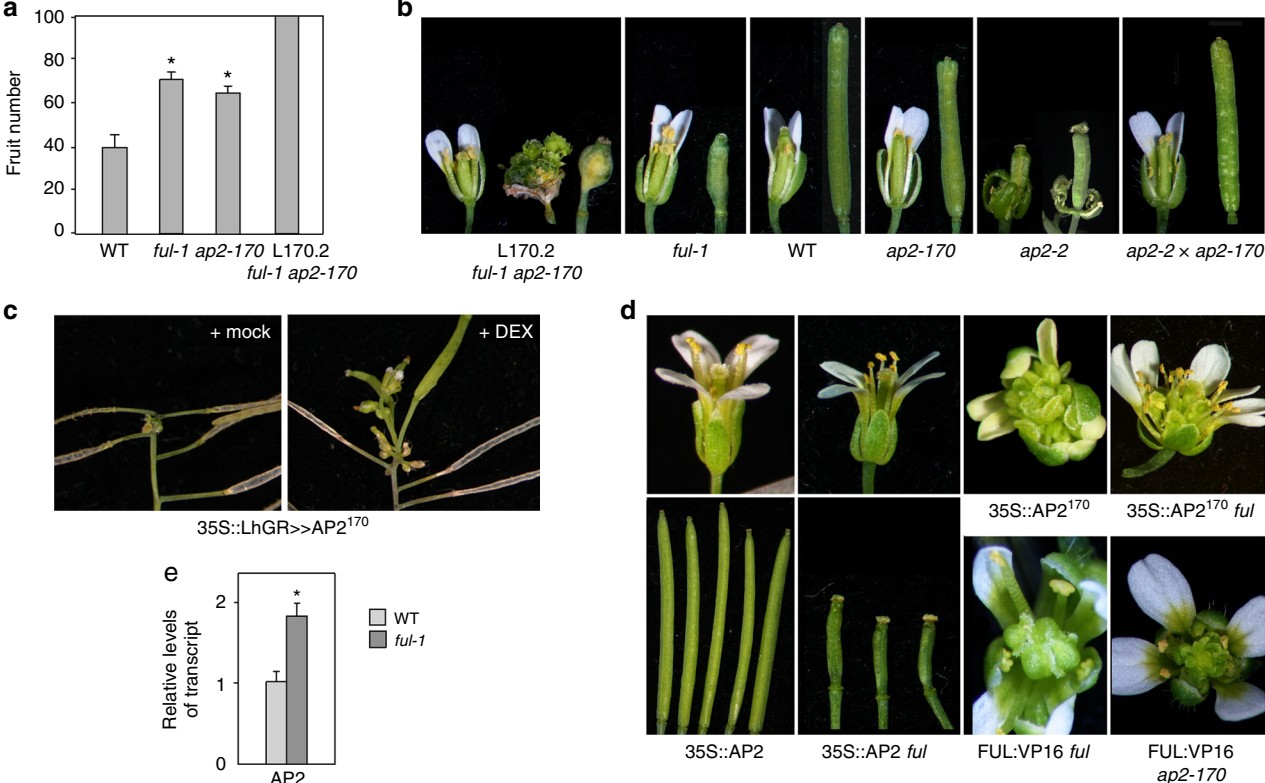

**Fig. 3** FUL controls GPA timing by regulating AP2 activity in the SAM. **a** L170.2, a mutant from a second-site mutagenesis on *ful-1*, shows a dramatically delayed GPA, with plants producing more than 100 fruits and usually dying of other causes before GPA can take place; L170.2 carries a mutation in the miR172 binding site of AP2 (renamed as *ap2-170*). When segregated from the *ful-1* background, the *ap2-170* mutation also delays GPA. $n = 10$ for all genotypes. **b** L170.2 bears highly indeterminate fruits, but not *ful-1* or *ap2-170* single mutants; *ap2-170* mutants or heterozygous *ap2-170/ap2-2* plants do not show the typical homeotic defects of *ap2-2* mutants. **c** Induction of AP2[170] expression in arrested plants reactivates SAM and flower and fruit production. **d** Overexpression of the AP2[170] allele causes highly indeterminate flowers both in WT and *ful* backgrounds, while overexpression of wild-type AP2 does not cause evident phenotypic defects in flowers or fruits. Similar indeterminate flowers to those produced by AP2[170] overexpression are produced by combining the *ap2-170* allele with a FUL:VP16 allele. FUL:VP16 also causes mild indetermination in fruits, especially in a *ful* mutant background. **e** Transcript levels of *AP2* are significantly elevated in the inflorescence meristems of *ful* mutants. Asterisks (*) indicate a significant difference ($P < 0.001$) from WT according to Student's *t* test

performed an independent ChIP-PCR analysis using the same transgenic lines, and detected clear enrichment for the fragments in the upstream regions of *AP2*, *SNZ*, *TOE1* and *TOE3*, but not in the regulatory region of *WUS* (Fig. 4a).

AP2 activity is regulated by miR172 at the post-transcriptional level, and AP2 has also been shown to feedback negatively on its own transcription[14, 21–23], complicating the extraction of significant conclusions from mRNA expression data that could reveal the nature of FUL-AP2 regulatory interaction. For these reasons, we took advantage of an enhancer trap line where a T-DNA containing a GUS reporter is inserted in the 5′UTR region of *AP2* (GT.1000845), expressing GUS under the control of regulatory sequences in the *AP2* genomic region and independent of miR172 regulation (Supplementary Figure 6a). GUS activity was detected at higher levels in *ful* mutants, indicating a role of FUL as a negative regulator of *AP2* promoter activity (Fig. 4b, Supplementary Figure 6b).

While these data suggest that negative regulation of *AP2* by FUL was independent of miR172, the elevated levels of *AP2* in *ful* meristems could still have at least two explanations: (i) FUL could activate miR172 transcription, leading to reduced *AP2* levels as has been described in fruits[24]; or (ii) FUL could be repressing *AP2* transcription directly. We generated FUL:VP16 lines consisting of a translational fusion of FUL with the strong transcriptional activation domain of the herpes virus protein VP16 under the

control of *FUL* regulatory sequences[25, 26] (Supplementary Figure 7a). If FUL activates the transcription of miR172, FUL: VP16 would result in higher levels of miR172 and hence a reduction in AP2 activity, while if FUL acted to repress *AP2* transcription, the FUL:VP16 protein would lead to higher expression of *AP2* and therefore we would expect indeterminacy defects. We observed that, in the FUL:VP16 background, the expression of the enhancer trap reporter inserted in the 5′UTR of *AP2* was increased (Fig. 3b), favoring the proposed role of FUL as an *AP2* repressor. Unfortunately FUL:VP16 lines could not be characterized for GPA defects because they showed strong ectopic *LFY* activation in the SAM that led to its conversion into a floral meristem (Supplementary Figure 7b, c), in accordance to previous reports of FUL being a *LFY* activator[27]. However, we did observe frequent formation of three-carpel gynoecia in FUL:VP16 flowers (around 20% of the pistils), that was reverted by *AP2* loss of function in a FUL:VP16 *ap2* background, again supporting the second scenario (FUL repressing *AP2* directly) (Supplementary Figure 7b). Moreover, *ful* FUL:VP16 flowers showed enhanced indeterminacy defects with extended *WUS* expression as expected if FUL and FUL:VP16 proteins were competing for targets and producing opposite effects on them (Fig. 3d, Supplementary Figure 7d). Finally, FUL:VP16 *ap2-170* plants produced highly indeterminate flowers consisting of several whorls of stamens and carpels, almost identical to those produced by previously

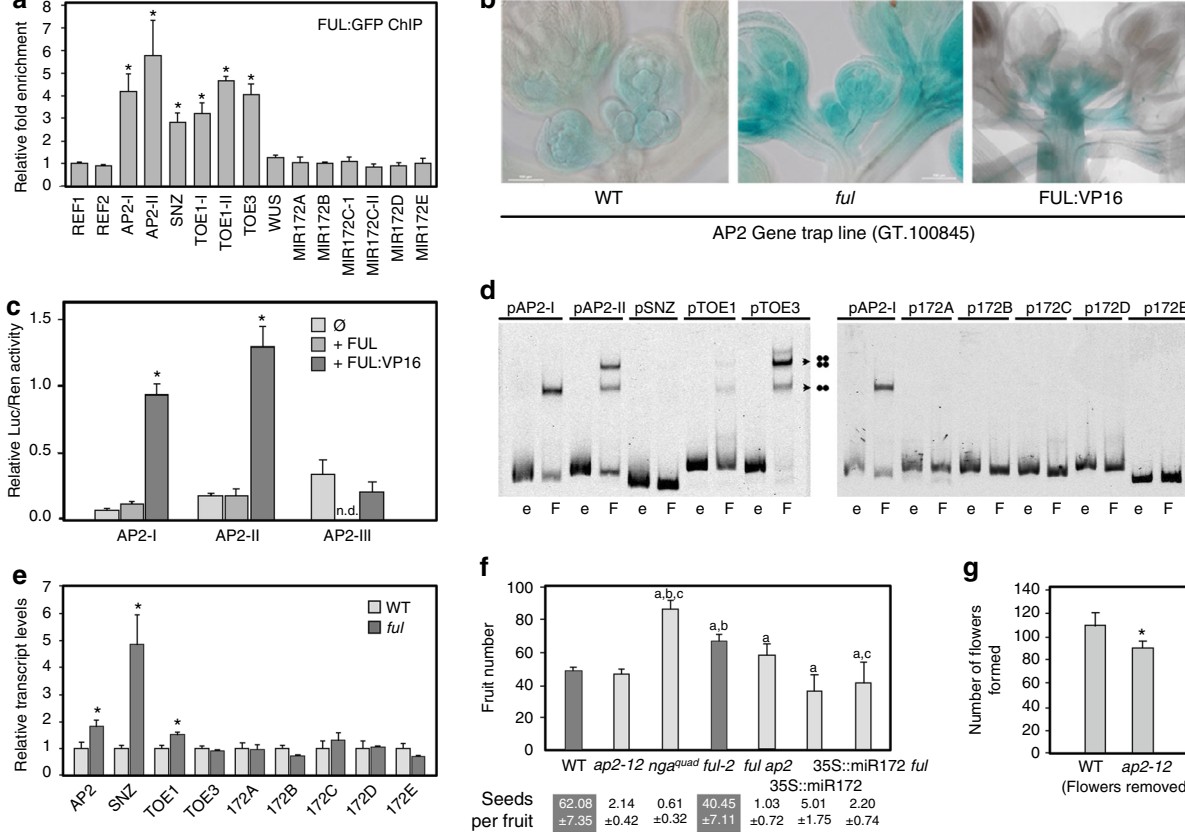

**Fig. 4** *AP2* and *AP2*-like genes control GPA timing. **a** ChIP-qPCR experiment showing the enrichment of the different fragments relative to *REF1* in ChIP samples from IM tissue of *ap1 cal* pFUL::FUL:GFP plants. Fragment enrichment was calculated as a percentage of the corresponding input sample. See Supplementary Data 1 for information about the fragments. Enrichment is detected for *AP2*, *SNZ*, *TOE1* and *TOE3*, but not for *WUS* and the *MIR172* genes. **b** In the gene trap line GT.100845, the activity of a miR-independent GUS reporter inserted in the 5'-UTR of the *AP2* gene is elevated both in *ful* and FUL:VP16 backgrounds. All panels correspond to heterozygotes AP2 (WT)/AP2-GT.100845. **c** Luciferase reporter assay in *N. benthamiana* leaves co-transformed with 35S::FUL or 35S::FUL:VP16 and different reporter lines where the AP2 promoter fragments enriched in FUL ChIP-seq results (AP2-I and AP2-II) and other non-enriched fragment (AP2-III) drive the expression of the *Luc* gene. Ren expression is used as internal control and Luc/Ren ratios are represented. **d** Binding of FUL to different DNA probes in an EMSA assay (as indicated on top of the lanes). The absence or presence of FUL (e, empty or F, FUL) is indicated at the bottom. The AP2-I fragment was loaded on both gels. The arrows indicate the shifts that can be attributed to binding of a FUL homodimer or a FUL homotetramer. More information about the probes can be found in Supplementary Data 1. **e** Transcript levels of AP2-like genes bound by FUL in inflorescence apices of wild-type and *ful* mutants: *AP2*, *SNZ* and *TOE1* are significantly overexpressed in *ful*, while transcript levels of *MIR172* genes are not affected. **f** *ap2-12* mutants produce a similar number of flowers than WT in spite of severely reduced fertility (noted by seed number per fruit), but much less than the unrelated sterile *nga^quad* mutant; 35S::miR172 plants, where expression of *AP2* and related genes is reduced, show further reduced flower production. The delayed GPA observed in *ful* mutants is not fully suppressed by the *ap2-12* mutation, but almost completely in the 35S::miR172 background. Shaded bars and shaded boxes in seed numbers indicate genotypes where fertility is above the threshold affecting GPA timing. Superscript letters indicate a significant difference (*P* < 0.001) from Col (**a**), *ap2-12* (**b**) and *ful-2* (**c**) controls, respectively, according to *t* test. **g** Induced sterility by continuous pruning of developing flowers causes increased flower production in WT and *ap2-2* mutants. Error bars represent the SE of four biological replicas in the case of expression analyses, three replicas for the ChIP-qPCR and six replicas for the Luc assays. Significant differences from the control (*t* test, *P* < 0.05) are indicated with an asterisk in **a**, **c**, **e**, **g**). *n* = 10 for all genotypes in **f**, **g**

described *AP2* alleles completely resistant to miR172 cleavage[17, 21] (Fig. 3d). In addition, we also tested the ability of FUL to act on the *AP2* promoter in vivo by luciferase transient reporter assays in *Nicotiana benthamiana* leaves. We generated reporter constructs in which three different fragments of the *AP2* promoter were cloned to direct reporter expression: two corresponding to the regions covered by the peaks enriched in the FUL ChIP-seq experiment (AP2-I and AP2-II, Supplementary Figure 5b) and the other in a region showing no enrichment (AP2-III, Supplementary Figure 5b). We observed that transiently expressed FUL protein was not able to activate any of the reporter constructs, as could be expected if FUL acted as a repressor, but transiently expressed FUL:VP16 activated the transcription of the two reporters driven by the promoter regions enriched in the

ChIP-seq experiments but not of that regulated by the non-enriched fragment (Fig. 4c).

To further support our hypothesis that FUL regulates *AP2* directly and not via miR172, we searched the same ChIP-seq results (GEO accession number GSE108455) for binding events of FUL in the miR172 precursors *MIR172A*, *MIR172B*, *MIR172C*, *MIR172D*, and *MIR172E*. Enrichment was not detected for any of the miR172 precursor loci in the ChIP-seq data, nor in an independent ChIP-PCR experiment that also included the *MIR172C* fragment previously reported to be bound by FUL in carpels[24] (Fig. 4a). We also used electrophoretic mobility shift assays (EMSA) to test the binding of FUL to the enriched sequences in the upstream regions of *AP2* and its close homologs, as well as to different *MIR172* fragments (including the fragment

reported[24]), and only observed shifts for *AP2*, *TOE1* and *TOE3*, but not for the *MIR172* fragments (Fig. 4d). In agreement with these data, *MIR172* precursor transcript levels were unchanged in *ful* SAM compared to the wild-type, while the expression of *AP2*, *TOE1* and *SNZ* was significantly upregulated (Figs. 3e, 4e). We should note that we could not detect binding of FUL to the *SNZ* promoter sequence tested in the EMSA assay, suggesting that FUL may require additional interacting factors to bind *SNZ* promoter in vivo. Taking all these data together, we found strong evidence that FUL acts as a direct repressor of *AP2* expression.

***AP2* and other miR172-regulated *AP2*-like genes regulate GPA.** If the extended activity of WUS in the SAM of *ful* mutants was due to the prolonged activity of AP2, a known positive regulator of *WUS* expression, *ap2* mutants should also show defects in GPA timing. Indeed, flower production in the loss-of-function *ap2-12* mutants was much lower than in other similarly sterile mutants such as *nga1 nga2 nga3 nga4* (*nga^{quad}*)[28] (Fig. 4f). Moreover, when flowers were manually removed in both wild-type and *ap2-12* mutants, GPA occurred significantly earlier in the *ap2* mutant background (Fig. 4g). Altogether, these results indicate that *AP2* loss-of-function caused early SAM termination.

However, *ful* defects in GPA timing were only weakly rescued by an *ap2* null mutation, suggesting that GPA control by FUL occurs not exclusively through *AP2* (Fig. 4f). Our identification of the other miR172-regulated *AP2*-like genes *TOE1*, *TOE3* and *SNZ* as direct targets of FUL (Fig. 4a, Supplementary Figure 5), together with the clearly increased expression of *SNZ* in the inflorescence apices of *ful* mutants (Fig. 4e), indicate that miR172 regulated *AP2*-like genes could also act downstream of FUL to control GPA. Supporting this idea, 35S::MIR172 plants, where the transcript levels of all members of the clade are reduced[22, 23], showed early GPA in spite of very reduced fertility, and when 35S::MIR172 was introgressed into *ful* mutants, it largely suppressed the delayed GPA in *ful* (Fig. 4f).

## Discussion

Our results demonstrate that *FUL* controls GPA timing through regulation of *AP2* and *AP2*-related genes, which in turn likely control, directly or indirectly, *WUS* temporal maintenance in the SAM.

Both *FUL* and *AP2*-like genes have been described as targets of the age-dependent pathway that controls developmental phase-transitions in *Arabidopsis*[29, 30]. This pathway involves miR156, whose levels decrease with plant age[29, 30]. miR156 negatively targets members of the SPL family of transcription factors, which in turn upregulate miR172 and also other genes involved in floral transition among which, interestingly, *FUL* has been identified[31]. Published reports on expression dynamics of SPL factors and miR172 accumulation in the SAM only span the period comprised between germination and floral transition[29, 30], and therefore it cannot be assumed that the antagonistic interaction of miR156 and SPLs that leads to gradually elevated levels of *FUL* and miR172 is maintained in the reproductive phase. However, in a recent study where the transcriptome of proliferative SAMs at early time points in inflorescence development were compared with the transcriptome of SAMs that had undergone GPA (GEO accession number GSE74386)[32], a threefold upregulation of *FUL* expression was observed in arrested meristems, suggesting that, at least, *FUL* accumulates through inflorescence development in the SAM as the plant ages.

Taking all this evidence together, we can propose a model for age-control of SAM fate and GPA timing in *Arabidopsis* that explains well the observed phenotypes (Fig. 5) and that would act

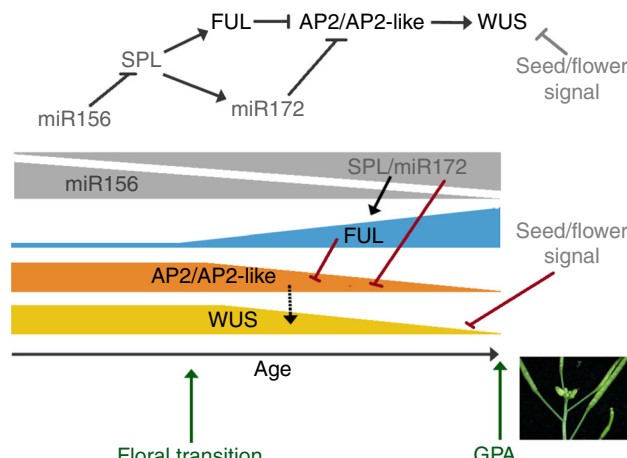

**Fig. 5** Model for temporal regulation of inflorescence meristem maintenance. WUS levels in the inflorescence SAM decrease with plant age. *WUS* expression is under positive regulation of AP2 and AP2-like factors in the SAM and negative regulation of an unknown signal from developing flowers/fruits. It has been previously shown that during the plant's life cycle, levels of miR156 decrease with age, concomitantly increasing the expression of members of the SPL family and of miR172. Hence, *AP2* levels in the SAM of the inflorescence would decrease with time by the combined repressing action of miR172 and FUL, which is strongly upregulated by SPL factors after floral transition. The decrease in *AP2* levels and the increasing repressing signal derived from fruits cause *WUS* to eventually turn off in the SAM. In *ful* mutants or in the *ap2-170* mutants, *AP2* and *WUS* are expressed in the SAM much longer, and therefore, GPA is delayed, while in loss-of-function *ap2/ap2*-like mutants, GPA occurs earlier. In sterile mutants, while the FUL-AP2 module should not change with age, the repressing signal from fruits is not active and *WUS* remains to be expressed longer

in parallel with the signals derived from flowers/seeds[1]. We suggest that *WUS* levels in the SAM partly depend on the activity of AP2 and AP2-related genes. Since FUL and miR172 are negative regulators of *AP2*, the increase in FUL and possibly in miR172 results in a decrease of AP2 activity and thus of *WUS* expression, eventually causing GPA (Fig. 5). This model is consistent with the observation that in *ful* mutants, where *AP2* regulation is impaired, or in the *ap2-170* allele, where miR172 suppression of AP2 activity is defective, GPA is significantly delayed, as well as with the early GPA observed in loss-of-function mutants in *AP2* and *AP2-like* genes.

Since GPA appears to be a general phenomenon in monocarpic species, it will be interesting to determine if the FUL/AP2/WUS pathway is conserved, particularly in crops of huge economic importance such as grain legumes or cereals. miR172/miR156 have been shown to control age-dependent processes in several species[33, 34]. While there are no reports so far linking *FUL* orthologs with the control of life span in other species, *ful* mutants in some *Arabidopsis* mutant backgrounds (i.e. *ful soc1* mutants) have been related to perenniality, a change from monocarpic to polycarpic habit[35, 36]. Moreover, FUL regulates *FLOWERING LOCUS C* in *Arabidopsis*[35], whose ortholog in the close relative *Arabis alpina*, *PERPETUAL FLOWERING 1*, regulates perennial flowering[37]. These data suggest a possible general role of this route in the control of longevity.

In summary, we have identified an FUL/AP2 module that regulates the mode and timing of GPA in *Arabidopsis*. This module would be under the control of age-dependent cues, acting in parallel with seed-dependent factors to regulate the life span and therefore fruit production.

## Methods

**Plant material and growth conditions.** *Arabidopsis thaliana* plants were grown in cabinets at 21 °C under LD (16 h light) conditions, illuminated by cool-white fluorescent lamps (150 $\mu$E/m$^2$/s$^1$), in a 1:1:1 by vol. mixture of sphagnum:perlite: vermiculite. To promote germination, seeds were stratified on soil at 4 °C for 3 days in the dark. The *Arabidopsis* plants used in this work were in the Col-0 background, except *ful-1*, *ap2-2*, L170.2 and the GT_5.100845 line that were in L*er*. Mutant alleles and transgenic lines have been previously described: *ful-1*[4], *ful-2*[27], *hec3* (SALK_005294 line)[6], *ap2-2*[38], *ap2-12*[39], *nga1-4 nga2-2 nga3-3 nga4-1 (nga-quad)*[28], WUS::GUS[10], CLV3::GUS[11], STM::GUS[12] and 35S::miR172[39].

35S::AP2 and 35S::AP2[170] were generated by cloning the two different AP2 CDS into the pEarley100 vector[40]. The FULp::FUL:VP16 transgene was generated by cloning a genomic fragment of the *FUL* locus, including 3.9 Kb of genomic sequence upstream the START codon, all exons and introns, fused in frame at the 3' end with the coding sequence of the strong activation domain of VP16, into the pBIN19 vector[40]. The 35S::LhG4:GR»AP2[170] construct was generated by cloning the AP2[170] CDS via a Gateway LR clonase II recombination reaction downstream of the artificial pOp6 promoter into the pOpOn2.1 binary vector derived from the pOpOff2 vector[41].

In all cases, *Agrobacterium* strain C58 pM090 was used to transform *Arabidopsis* using the floral dip protocol[42], and transgenic lines carrying a single transgene insertion were selected. For lines transformed with 35S::AP2 or 35S:: AP2[170], expression of the transgenes was confirmed by semi-quantitative RT-PCR of AP2 performed on cDNA extracted from seedlings belonging to seven random independent T2 families (Supplementary Figure 8).

Primer sequences used for cloning and semiquantitative RT-PCR are detailed in Supplementary Data 1.

**Flower and fruit number quantification.** Plants were grown as described above. Elongated fruits (or flowers after anthesis in sterile backgrounds) were quantified in the main inflorescence for at least ten plants of each genotype. Plants showing obvious health problems or off-looking were discarded. Experiments were replicated independently twice, obtaining comparable results, although only one experiment is represented in each figure. For experiments involving manual pruning to avoid seed formation, flowers at the anthesis stage were removed every 2 days with clean tweezers until GPA or terminal structure occurred and number of nodes producing flowers were quantified. For seed quantification, fruits from nodes 6 to 10 in the main inflorescence of each plant ($n = 10$) were collected, and seeds were counted for each individual fruit ($n = 50$).

**Positional cloning.** To positionally clone the gene affected by the L170.2 mutation, we outcrossed *L170.2* (in a *ful-1* L*er* genetic background) to *ful-2* (in Col-0). Mapping of the *L170.2* mutation was performed as described in refs. [43, 44]. In brief, for low-resolution mapping, the DNA of 50 F$_2$ phenotypically mutant plants was individually extracted and used as a template to multiplex PCR co-amplify 32 SSLP and In/Del molecular markers using fluorescently labeled oligonucleotides as primers. For fine mapping, 98 additional F$_2$ plants were used to iteratively assess linkage between *L170.2* and molecular markers designed according to the polymorphisms between L*er* and Col-0 described at the Monsanto Arabidopsis Polymorphism Collection database (http://www.arabidopsis.org). Linkage analysis of the F$_2$ mapping populations obtained allowed the IB-UMH Gene Mapping Facility (Elche, Spain) to delimit a candidate interval of 99 kb in chromosome 4, between markers cer448356 and cer448311, containing 22 annotated genes. Sequencing of the genomic region in this interval of L170.2 mutants identified a single polymorphism in the AP2 CDS.

**Chromatin immunoprecipitation.** A pFUL::FUL-GFP[45] construct was introduced in the *ap1 cal AP1-GR* background[46] and non-induced plants were grown for 6 weeks, after which all inflorescence meristems were harvested. Tissue fixation and immunoprecipitation with the GFP antibody were performed as described[47–50]. In short, approximately 1 g of tissue was harvested per sample, grinded and resuspended in MC buffer (10 mM sodium phosphate, pH 7.0). The tissue was fixed for 2× 15 min in MC buffer with 1% formaldehyde under vacuum and nuclei were isolated. The chromatin was then fractionated and solubilized on ice using a probe sonicator. The sonicated material was centrifuged and the supernatant used for the IP. At this point, 10% of the sample was set aside to serve as the input control. The IP was performed using $\mu$MACS anti-GFP microbeads (Miltenyi Biotec). Fifty microliters of anti-GFP beads were added to each sample, and the samples were incubated for 1 h on a rotating wheel at 4 °C, followed by immobilization and washing of the beads on the $\mu$-column using the $\mu$MACS separator. Elution of the captured chromatin was achieved by adding three times 50 $\mu$l hot elution buffer (1% SDS, 50 mM Tris-HCl pH8, 10 mM EDTA, 50 mM DTT, 95 °C) to the beads. All samples (including the input samples) were reverse cross-linked using proteinase K, and purified using EtOH precipitation and column purification. For the ChIP-qPCR, three biological replicas were compared with the corresponding input samples. The fragment amplified from *WUS* promoter corresponded to a peak weakly enriched in the ChIP-seq experiment (below the significance threshold). Primer sequences used for the ChIP-PCR are detailed in Supplementary Data 1.

The ChIP-seq data (deposited by Hilda van Mourik, Jose M. Muiño, Cezary D. Smaczniak, Marian Bemer, Dijun Chen, Gerco C. Angenent, and Kerstin Kaufmann) were retrieved from the NCBI Gene Expression Omnibus (GEO) under accession number GSE108455[51].

**Luciferase activity assay.** Genomic regions identified as binding sites for FUL by ChIP-seq (fragment AP2-I and AP2-II) were amplified by PCR, as well as one adjacent region (fragment AP2-III), and cloned in the pUPD2 vector (https:// gbcloning.upv.es/tools/). To generate reporters, the different fragments were combined by Golden Braid technology with a minimal 35S promoter directing the LUC gene expression in a pDGB3$\alpha$1 vector. The different $\alpha$1 vectors produced were then combined with an pDGB3$\alpha$2 vector containing a 35S::REN and a 35S:: p19 inhibitor in a pDGB3 $\Omega$1 vector (https://gbcloning.upv.es/tools/). The different $\Omega$ vectors obtained were agroinfiltrated alone or co-agroinfiltrated with 35S::FUL:: VP16 or 35S::FUL. The Luciferase expression assays were performed by transient transformation of *N. benthamiana* leaves by Agrobacterium infiltration, which was performed as previously described[52] with minor modifications[53]. Briefly, 300 $\mu$l of *Agrobacterium* containing the reporter and or effector plasmids was infiltrated into a young *Nicotiana* leaf at three points. Firefly luciferase and Renilla luciferase were assayed 3 days after infiltration using the Dual-Luciferase Reporter Assay System (Promega). Data were represented as the ratio of LUC/REN. Background controls were obtained by infiltrating only the reporter construct. At least three plants at the same developmental stage were used for each treatment, and the experiments were repeated six times.

**Electrophoretic mobility shift assay.** The complete *FUL* coding sequence was amplified from cDNA and cloned into the pSPUTK vector. The pSPUTK vector allowed in vitro protein synthesis using the TnT® SP6 High-Yield Wheat Germ Protein Expression System (Promega) according to the manufacturer's instructions. All promoter fragments were amplified from genomic DNA and cloned into pGEM-T (Promega). EMSAs were performed essentially as follows[54]: Oligonucleotides were fluorescently labeled using DY-682. Labeling was performed by PCR using pGEM-T-specific DY-682-labeled primers followed by PCR clean-up. Binding of the protein complexes to the labeled DNA was performed for 1 h on ice using binding reaction mixture (1.2 mM EDTA (pH 8.0), 0.25 mg/ml BSA, 7.2 mM Hepes (pH 7.3), 0.7 mM DTT, 60 $\mu$g/ml salmon sperm DNA, 1.3 mM spermidine, 2.5% CHAPS, 8% glycerol, 3.3 nmol/ml double-labeled dsDNA, and 2 $\mu$l of in vitro-synthesized proteins). The entire binding mixture was then loaded on a 5% polyacrylamide gel. The gel was run in 1× TBE buffer (10× TBE per liter: 10.8 g Tris base, 5.5 g Boric acid, 7.4 g EDTA) at 75 V for about 90 min. Gel-shifts were visualized using a LiCor Odyssey imaging system at 700 nm. Sequences of primers can be found in Supplementary Data 1.

**β-Glucuronidase staining.** For β-Glucuronidase (GUS) histochemical detection, samples were treated for 15 min in 90% ice-cold acetone and then washed for 5 min with washing buffer (25 mM sodium phosphate, 5 mM ferrocyanide, 5 mM ferricyanide, and 1% Triton X-100) and incubated from 4 to 16 h at 37 °C with staining buffer (washing buffer + 1 mM X-Gluc). Following staining, plant material was fixed, cleared in chloral hydrate, or included in paraffin and sectioned. Samples were mounted to be viewed under bright-field microscopy.

**Quantitative RT-PCR.** Inflorescence meristems were harvested and trimmed under the stereo microscope until no buds older than stages 3–4 were visible anymore. Four biological replicas were sampled, each containing about 20 inflorescence meristems. RNA was extracted using CTAB/LiCL, DNase treated with Turbo DNase (Ambion AM1907), and 1 $\mu$g of RNA was used for the cDNA preparation with Superscript II (Invitrogen). The qPCR was performed on the iCycler iQ5 system using iQ SybrGreen (BioRad). The following program was used: 3 min 95 °C, 40 cycles (10 s 95 °C, 45 s 60 °C).

**Statistical methods.** Two-tailed Student's *t*-test was performed whenever two groups were compared. Statistical significance was determined at $P < 0.001$ unless otherwise indicated.

**Scanning electron microscopy.** Five inflorescences for each condition were fixed overnight at 4 °C in FAE solution (ethanol:acetic acid:formaldehyde:water, 50:5:3.5:41.5, v/v/v/v), dehydrated through an ethanol series and critical-point dried in liquid $CO_2$. Samples were coated with gold-palladium (4:1) and observed in a Ultra55 Zeiss electronic microscope, working at 2–4 kV and a scanning speed of 200 s per image.

**In situ hybridization.** In situ hybridizations were performed as described[27, 55]. For hybridization in sections, tissue was fixed for 2 h in FAE solution, dehydrated, embedded and sectioned to 8 $\mu$m. After dewaxing in histoclear and rehydrating, sections were treated for 20 min in 0.2 M HCl, neutralized for 10 min in 2× SSC and then incubated for 30 min with 1 $\mu$g/ml Proteinase K at 37 °C. Proteinase action was blocked by treating with 2 mg/ml Gly for 5 min and postfixation in 4% formaldehyde for 10 min. Subsequently, sections were dehydrated through an

ethanol series before applying the hybridization solution (100 μg/ml tRNA; 6× SSC; 3% SDS; 50% formamide, containing approx. 100 ng/μl of antisense DIG-labeled RNA probe), and left overnight at 52 °C. Then, sections were washed twice for 90 min in 2× SSC: formamide (50:50) at 52 °C before performing the antibody incubation and color detection. For whole mount hybridizations, samples were fixed in FAE solution for 1 h and dehydrated through an ethanol series. For permeabilization, samples were incubated in histoclear:methanol 50:50, cell wall digested for 6 min, postfixed in 4% formaldehyde for 15 min and then incubated for 15 min with 80 μg/ml Proteinase K at 37 °C, followed by a second postfixation in 4% formaldehyde for 30 min. Subsequently, samples were incubated overnight in hybridization solution (100 μg/ml tRNA; 100 μg/ml salmon sperm; 5× SSC; 0,1% Tween20; 0,2X Denhardt's solution: 50% formamide, containing approx. 100 ng/μl of antisense DIG-labeled RNA probe) at 50 °C.

Finally, samples were washed twice for 90 min in 2× SSC: formamide (50:50) at 50 °C before performing the antibody incubation and color detection.

At least five samples for each condition were observed. LFY and WUS probes were prepared from the full length CDSs of both genes.

**Data availability**. The authors declare that all data supporting the findings of this study are available within the manuscript and its supplementary files or are available from the corresponding author on request.

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

## Acknowledgements

We thank M. Schmid and D. Weigel (MPI, Tubingen) for sharing materials and M. Blazquez, F. Madueño (IBMCP), and J.J. Ripoll (UCSD), for helpful discussions. We also thank M.R. Ponce and J. L. Micol (Instituto de Bioingeniería, Universidad Miguel Hernández de Elche) for their help with the positional cloning of the AP2 gene and H. van Mourik (Wageningen University) for help with the ChIP-seq analysis. Our work was supported by grants from the Spanish DGI to C.F. (CSD2007-00057-B, BIO2012-32902, BIO2015-64531-R). V.B. was supported by a predoctoral fellowship of the Generalitat Valenciana.

## Author contributions

V.B. and C.F. conceived and designed the study and wrote the manuscript together with M.B., V.B., I.M.-F., M.B., S.S., and C.F. performed the genetic, physiological and molecular work. M.B. and K.K. produced the ChIP-seq data. M.F.Y., G.C.A. and C.F. provided scientific guidance throughout the project. All authors discussed the results and commented on the manuscript.

## Additional information

**Competing financial interests:** The authors declare no competing financial interests.

