## [Peer Review File · Nature Communications]

Reviewers' comments:

Reviewer #1 (Remarks to the Author):

The present work by Balanza and colleagues, studies the genetic control of proliferative arrest in inflorescence meristems. The authors base their research on the observation that plants lacking the activity of the MADS Box transcription factor FRUITFUL (FUL) produce more flowers than wild type plants. Therefore, FUL has a negative role in meristem maintenance. That is actually not surprising since there is an important body of literature linking FUL to meristem determinacy and cell differentiation, in which some of the senior authors of the present manuscript have participated. To rule out that the increased number of flowers produced in ful mutants is a by side consequence of reduce seed setting, they compare seed and flower production in those and other mutants impaired in reproduction. Likewise, they compared flower production between wild type and ful mutants in which flowers were removed to prevent fertilization. Later on, the authors analyse the expression of the essential factor for meristem maintenance WUS using a WUS::GUS reporter line. WUS expression declines over time but faster in WT than in ful plants. The authors need to complement this approach with an additional method that GUS staining, preferentially with RNA in situ experiments and QRT-PCR to quantify WUS levels. Next, the authors mention a ChIP-seq analysis to determine whether FUL can bind directly WUS regulatory sequences.

To try to elucidate the mechanism by which FUL might be contributing to WUS regulation, the authors performed an enhancer suppressor screening on ful mutant plants. At least one of the obtained lines showed extended life span of the SAM. The mutant gene underlying such feature was found to be AP2. AP2 had been formerly isolated in through a similar strategy but using a ful rpl sensitized background. In addition, AP2 has been shown to regulate WUS expression in former studies. The new ap2-170 allele found, carries a aminoacid substitution at the miR172 target site. The ap2-170 allele is a gain of function derived from escaping miR regulation or because of the aminoacid substitution as corroborated by the meristem reactivation observed in senescent plants when AP2-170 was induced. As formerly suggested in Ripoll et al., (2011), the authors place AP2 function downstream of FUL by showing that just the overexpression of the new AP2 allele is able to affect organ determinacy in ful mutant backgrounds. Next, the authors found that AP2 mRNA levels were higher in ful mutants when compared to WT, suggesting FUL represses AP2. To gain further insights into the FUL-AP2 module, the authors used a reporter line in which a GUS happened to be inserted at the 5' UTR of AP2. According to the authors, that higher GUS levels were observed in ful mutants is a probe that FUL is regulating AP2 expression independently of its miR172 regulator. Authors should check AP2 transcript levels in those backgrounds to make sure there is no AP2 feedback through binding to its own promoter. That is especially critical since the authors rule out the role of FUL on miR172 in AP2 regulation. In this line, the authors could use miR172 and MIM172 overexpressors to address the role of miR172 in mediating FUL regulation on AP2 levels, material the authors have used in other publications from their group. The authors could find signatures of FUL binding to the AP2 promoter and the ones from other members of the AP2-like clade under miR172 regulation in ChIP-seq experiments. Nevertheless, the information about the number of biological and technical replicates used in those studies is not available in the text. Additionally, to corroborate by Q-PCR the results obtained by ChIP-seq, the authors used a line overexpressing FUL which is different from the one used for the ChIP-seq approach. Authors need to repeat that control using the very same background that for ChIP-seq. The authors show that mutants impaired in the function of those AP1-like genes result also in altered GPA timing.

Although the paper is of interest, the results presented are not surprising. In addition, the authors need to further define the mechanism through which FUL is regulating AP2 levels, either transcriptionally or through miR172 regulation.

Reviewer #2 (Remarks to the Author):

This study implicates FRUITFULL (FUL), a MADS-BOX gene in determining the timing of shoot meristem arrest. *ful* mutant SAMs have been shown to produce higher number of flowers than that of wild type SAMs before terminating. They show that this is not due to lower seed set/sterility as suggested by the genetic analysis and periodic removal of flowers. At the molecular level, the study revealed that WUS expression is maintained over a prolonged period of time *ful* mutants suggesting that FUL function is required for repressing WUS expression. To identify the factors regulating the FUL-mediated repression of WUS, a genetic screen, was performed which identified an allele of *ap2* (*ap2-170*, a microRNA resistant version) as an enhancer of *ful-1* mutant. Then the authors went onto show that overexpression of this *ap2* mutant protein delayed the global proliferative arrest (GPA). The expression analysis showed that FUL functions as a AP2 repressor and the ChIP analysis showed that FUL protein directly bound AP2 promoter regions. Based on these observations, authors propose a model that age control/GPA signals act in parallel with flower/seed derived signals. The increase in FUL and miR172 levels cause downregulation of AP2 levels and subsequent decrease in WUS levels leading to the GPA. Overall the study is well done and addresses an intangible question of cellular/tissue aging. The study can be improved by considering the following points.

1. Global proliferative arrest (GPA) is yet to be described as a discrete and temporally resolved cellular events. Is GPA due to cell division arrest or differentiation of stem cells. A better quantification of cell division events and analysis of other stem cell specific (CLV3 for example and STM), differentiation markers and hormonal sensors such as pTCS (considering that cytokinin could be important) at different stages prior to the initiation of GPA is required. The current *pWUS:GUS* analysis is not sufficient (see point#2).
2. Continuing on my argument on the importance of taking a global view of tissue aging, instead the current study focuses too much on the timing of termination of one molecule that is WUSCHEL (WUS). First of all the images showing lack of WUS-GUS in (Fig. 1E) in wild type are not convincing because I do not see expression in floral meristems which are expected to express WUS as they are much younger than the SAM. Yes even if we assume that WUS downregulation occurs, that alone can not explain a universal phenomenon such as GPA. In fact *wus* mutants do produce several shoots that do not senesce for several months. Therefore, if FUL is acting through repression of AP2, analyzing AP2-mediated transcriptome may provide a global view of GPA. Is there anything known about AP2-regulated transcriptome or even the FUL-regulated transcriptome that may better explain involvement of FUL in GPA?
3. Molecular analysis showing that AP2 is directly repressed by FUL is interesting and important for understanding of the onset of GPA. However, I find that this analysis is incomplete as it is only based on ChIP expts. Authors must consider experiments that show direct binding (EMSA) and transient transfection assays to prove the point that AP2 is directly repressed by FUL.

Reviewer #3 (Remarks to the Author):

Balanza et al. described a series of genetic analysis of events leading to the global proliferative arrest, flowering and developing seeds. It is an open question how signals from developing seeds and age-dependent factors are integrated during the process. They at first observed the *ful* mutant and its delayed GPA. Then they screened an EMS enhancer-suppressor mutagenesis on *ful-1* and isolated a line of which nucleotide substitution was mapped in the last exon of the AP2 gene. The line gave the clue to link FUL-AP2-WUS and fill the gap between the GPA, flowering and setting seeds. The paper has an impact to shed light on FUL role in the flower/seed formation, thus we would like to know further information.

Major points:

The authors proposed a new model depicted in Fig.4. To reinforce the model, I dare to ask regarding to Figure 1. What happens if WUS::GUS patterns are checked in ap2-170, ful-1/ap2-170 background? I wondered if total seeds set in each line might be roughly constant, if products between number of mature fruits and seeds/fruit are calculated in Fig.1B. Any comment on this point? If the targeting of AP2 mRNA by miR172 is important, the authors could address the issue whether miR172-resistant AP2 gene could mirror the ap2-170 phenotype.

Minor points:

P.3, last two sentences could be modified to be reader-friendly
> ... when transformed with a 35S::AP2170 construct (Fig.2D), but not when transformed with 35S::AP2 (Fig S5),

Reviewer #4 (Remarks to the Author):

In this manuscript Balanza and colleagues describe a regulatory circuit that involves FRUITFULL (FUL) and APETALA2 (AP2) as regulator of global proliferative arrest (GPA) in *Arabidopsis thaliana*. AP2 has previously been implicated in meristem maintenance (i.e. Wuerschum et al., 2006) but the current study goes beyond this previous knowledge and provides information on the molecular regulation of AP2 by the MADS-domain transcription factor FUL, specifically during GPA. Given the GPA can affect yield in crops the findings should be of interest to a wider readership even though the current manuscript does not provide any evidence that the mechanism identified in *Arabidopsis* is conserved in other species.

According to the model provided by the authors AP2 integrates signals from FUL and miR172. One prediction from this model is that reduced miR172 levels should result in elevated AP2 and indirectly delay GPA. miR172 is encoded by several genes in *Arabidopsis thaliana*, so a detailed genetic analysis is not really possible. However, I think that knocking down miR172s by a mimicry approach could be very informative. The prediction would be that reduced miR172 (possibly by expressing MIM172 from the FUL promoter) in a ful mutant should further delay GPA.

The manuscript is well written but very concise. More details and background information should be provided, if space permits. Also, the figures not always fully support the claims made in text (see specific comments and questions provided below for details).

Specific comments and questions:

Introduction

- page 2, line 2: introduce the abbreviation GPA also at first use in the main text, not only in the abstract
- page 2, line 4: sentence seems to be incomplete. Maybe "... optimize allocation of resources..."

Results

- "ful mutants have delayed GPA", page 2:

The authors claim that the reduced seed set observed in ful mutants is not the cause of the GPA because two other mutants with reduced seed set, fer and hec3, do not show GPA. However, the number of different mutants examined (3) is rather small and I wonder how specific the GPA phenotype in ful really is.

- "WUS repression in the SAM correlates with GPA", page 3:

GUS reporter at best give semi-quantitative information about gene expression and while the data provided on WUS expression being maintained in ful mutants is most likely correct I would feel more comfortable if the authors could provide independent evidence for this. i.e. RNA in situ hybridization (which I realize are also semi-quantitative at best) or qRT-PCR (which could be tricky as WUS is also expressed in floral primordia)

- "FUL regulates GPA by fine tuning AP2 activity in the SAM"

page 3, line 4: "... screened an EMS enhancer-suppressor mutagenesis on ful-1." Should be changed to "... performed an EMS enhancer-suppressor mutagenesis screen on ful-1."

page 3, line 6: the subject seems to be missing in the sentence "In addition, showed a strong...". Possible "In addition, the mutant showed a strong..."

page 3, line 12: mutations in the miR172 binding site of AP2 (-like) transcription factors (at least when expressed at high levels) have been shown to delay flowering. Does the ap2-170 mutant flower later than wildtype? If it does, how might this be related to delayed GPA in ful?

page 3, line 13, 16: No conclusion on homeotic phenotypes in the outer three whorls possible based on the pictures provided in Fig 2B as this panel only shows fruits, not flowers. Please provide pictures of expanded flowers to support the claim that ap2-170 protein is largely functional.

page 3, line 21: The author write that "AP2 induction in senescent wildtype plants that already had undergone GPA restored the activity of the SAM..." How did the authors ensure that the SAM had already undergone GPA?

page 3, line 27: How did the authors confirm over-expression of AP2 in these lines? In particular in the 35S::AP2 line which apparently did not show any phenotypes.

page 4, line 4: The authors should provide expression data for other meristem-expressed genes, to show that the elevated expression was specific to AP2 and not caused by general differences in meristem size/architecture (which has been described for other Pap2 alleles; e.g. Wuerschum et al, 2006).

page 4, line 25: Fig 2F (?)

page 4, line 30: "...observe frequent formation of three-carpel gynoecia...". Can this phenotype be quantified?

page 4, line 33: Fig S7 is fragmented, uses different fonts, and is generally not very nice. Please reassemble figure

page 5, line 2: "...so we analysed our ChIP-seq experiment in FUL::FUL:GFP inflorescences...". Which ChIP-seq experiment is this? Please provide reference or experimental details.

page 5, line 6: "...while no significant FUL binding was observed in any of the MIR172 genes...". Even though this negative evidence the actual ChIP-seq data on miR172 not being bound by FUL should be included in Fig S8 to substantiate the claims made.

The same font should be used for figures throughout the manuscript (incl. supplementary material).

- "AP2 and other miR172-regulated AP2-like genes participate in GPA regulation"

page 5, line 7: "...GPA occurred significantly earlier in the ap2 ...". No significance level indicated in Fig 3B. Which statistical test was used?

page 5, line 16: Font color

Figure legends:

Fig 1A: "ful mutants produce more fruits than WT". This is actually not clearly visible from the figure. What can be seen is that the ful inflorescence is taller but the number of fruits is not clear.

Fig 2B: "ap2-170/ap2-2 plants do not show the typical homeotic defects of ap2-2 mutants". The pictures show only fruits but not whorls 1 and 2 which are normally strongly affected in ap2 mutants. Please provide pictures of whole open flowers (or avoid talking about "typical homeotic defects").

Reviewers' comments:

Reviewer #1 (Remarks to the Author):

The present work by Balanza and colleagues, studies the genetic control of proliferative arrest in inflorescence meristems. The authors base their research on the observation that plants lacking the activity of the MADS Box transcription factor FRUITFUL (FUL) produce more flowers than wild type plants. Therefore, FUL has a negative role in meristem maintenance. That is actually not surprising since there is an important body of literature linking FUL to meristem determinacy and cell differentiation, in which some of the senior authors of the present manuscript have participated. To rule out that the increased number of flowers produced in ful mutants is a by side consequence of reduce seed setting, they compare seed and flower production in those and other mutants impaired in reproduction. Likewise, they compared flower production between wild type and ful mutants in which flowers were removed to prevent fertilization. Later on, the authors analyse the expression of the essential factor for meristem maintenance WUS using a WUS::GUS reporter line. WUS expression declines over time but faster in WT than in ful plants. The authors need to complement this approach with an additional method that GUS staining, preferentially with RNA in situ experiments and QRT-PCR to quantify WUS levels.

Response: We thank the reviewer for the valuable comments and suggestions. In the revised manuscript we have provided new analyses of WUS expression by both RNA in situ hybridization and GUS reporter analyses in whole mounts. The results obtained are similar with both approaches and reinforce our conclusions.

Next, the authors mention a ChIP-seq analysis to determine whether FUL can bind directly WUS regulatory sequences.

To try to elucidate the mechanism by which FUL might be contributing to WUS regulation, the authors performed an enhancer suppressor screening on ful mutant plants. At least one of the obtained lines showed extended life span of the SAM. The mutant gene underlying such feature was found to be AP2. AP2 had been formerly isolated in through a similar strategy but using a ful rpl sensitized background. In addition, AP2 has been shown to regulate WUS expression in former studies. The new ap2-170 allele found, carries an amino acid substitution at the miR172 target site. The ap2-170 allele is a gain of function derived from escaping miR regulation or because of the amino acid substitution as corroborated by the meristem reactivation observed in senescent plants when AP2-170 was induced. As formerly suggested in Ripoll et al., (2011), the authors place AP2 function downstream of FUL by showing that just the overexpression of the new AP2 allele is able to affect organ determinacy in ful mutant backgrounds. Next, the authors found that AP2 mRNA levels were higher in ful mutants when compared to WT, suggesting FUL represses AP2. To gain further insights into the FUL-AP2 module, the authors used a reporter line in which a GUS happened to be inserted at the 5'UTR of AP2. According to the authors, that higher GUS levels were observed in ful mutants is a probe that FUL is regulating AP2 expression independently of its miR172 regulator. Authors should check AP2 transcript levels in those backgrounds to make sure there is no AP2 feedback through binding to its own promoter. That is especially critical since the authors rule out the role of FUL on miR172 in AP2 regulation.

Response: We would like to clarify that GUS detection was done in GT.100845 heterozygotes, since GT.100845 homozygotes are similar to *ap2* null mutants. Therefore, one functional copy of AP2 was present in all backgrounds (a sentence was added to the corresponding figure legend to indicate this). We agree that we cannot rule out completely that AP2 feeds back on its own promoter, but we have already shown that in *ful* mutant background, AP2 levels are elevated in the SAM (Fig 3e in the revised ms.), so we do not feel that quantification of AP2 mRNA levels in the *ful* GT.100845/+ background would add any new information. Interestingly, there are several reports in the literature that indicate that AP2 negatively regulates its own promoter (for example, Schwab et al, 2005, and Yant et al, 2010, refs 21 and 38 in the revised ms). Therefore, if any, the elevated AP2 mRNA levels in *ful* mutant background would cause a decrease in AP2 promoter activity, not the observed increase. In addition, the fact that GUS activity is **both elevated** in the *ful* GT.100845/+ and the FUL:VP16 GT.100845/+ background strongly favors the role of FUL as a direct repressor of AP2. Moreover, in the revised manuscript we have provided additional evidence supporting that FUL directly regulates AP2 promoter and not miR172 in the SAM (new ChIP and expression data, EMSAs and LUC transient activation assays). Altogether, we hope to have convinced the reviewer on this point.

In this line, the authors could use miR172 and MIM172 overexpressors to address the role of miR172 in mediating FUL regulation on AP2 levels, material the authors have used in other publications from their group.

Response: We believe that the new evidence added to the revised version of the manuscript strongly supports our original claim of FUL directly repressing AP2 transcription by binding to its promoter. We agree that we cannot completely rule out an additional role of miR172 regulation by FUL in the SAM, but our ChIP and expression data argue against a prominent role of a FUL direct regulation of miR172 in this process. We are not sure how miR172ox or MIM172ox could help to clarify this point, and would welcome suggestions in this direction.

The authors could find signatures of FUL binding to the AP2 promoter and the ones from other members of the AP2-like clade under miR172 regulation in ChIP-seq experiments. Nevertheless, the information about the number of biological and technical replicates used in those studies is not available in the text. Additionally, to corroborate by Q-PCR the results obtained by ChIP-seq, the authors used a line overexpressing FUL which is different from the one used for the ChIP-seq approach. Authors need to repeat that control using the very same background that for ChIP-seq. The authors show that mutants impaired in the function of those AP1-like genes result also in altered GPA timing.

Response: Thanks for pointing this out. We have included new information and the suggested experiments in the revised version (see figure 4 and M&M section in Supp. material)

Although the paper is of interest, the results presented are not surprising. In addition, the authors need to further define the mechanism through which FUL is regulating AP2 levels, either transcriptionally or through miR172 regulation.

Response: We thank the reviewer for appreciating the interest of the study. However, we do believe that our results are novel. First, and more importantly, we characterize for the first time a genetic pathway controlling GPA timing, which so far had only been the subject of descriptive studies and only conclusively related to plant fertility. In addition, we uncover a previously unknown role for FUL in controlling GPA. Also, the reported regulatory nature of FUL-AP2 interaction is also novel, since previously it had only been shown that FUL could regulate AP2 through miR172 activation and we report that FUL in the SAM regulates AP2 directly. We believe that this study will serve as starting point to investigate the mechanisms controlling GPA in monocarpic plants in a much wider context.

Reviewer #2 (Remarks to the Author):

This study implicates FRUITFULL (FUL), a MADS-BOX gene in determining the timing of shoot meristem arrest. *ful* mutant SAMs have been shown to produce higher number of flowers than that of wild type SAMs before terminating. They show that this is not due to lower seed set/sterility as suggested by the genetic analysis and periodic removal of flowers. At the molecular level, the study revealed that WUS expression is maintained over a prolonged period of time *ful* mutants suggesting that FUL function is required for repressing WUS expression. To identify the factors regulating the FUL-mediated repression of WUS, a genetic screen, was performed which identified an allele of *ap2* (*ap2-170*, a microRNA resistant version) as an enhancer of *ful-1* mutant. Then the authors went onto show that overexpression of this *ap2* mutant protein delayed the global proliferative arrest (GPA). The expression analysis showed that FUL functions as a AP2 repressor and the ChIP analysis showed that FUL protein directly bound AP2 promoter regions. Based on these observations, authors propose a model that age control/GPA signals act in parallel with flower/seed derived signals. The increase in FUL and miR172 levels cause downregulation of AP2 levels and subsequent decrease in WUS levels leading to the GPA. Overall the study is well done and addresses an intangible question of cellular/tissue aging.

Response: We thank the reviewer for appreciating the interest and importance of the study and for comments and suggestions, which we try to address below.

The study can be improved by considering the following points.

1. Global proliferative arrest (GPA) is yet to be described as a discrete and temporally resolved cellular events. Is GPA due to cell division arrest or differentiation of stem cells. A better quantification of cell division events and analysis of other stem cell specific (*CLV3* for example and *STM*), differentiation markers and hormonal sensors such as pTCS (considering that cytokinin could be important) at different stages prior to the initiation of GPA is required. The current pWUS:GUS analysis is not sufficient (see point#2).

Response: We agree with the reviewer that we still have to characterize in much more depth the GPA process and related events, and that doing this would be highly important. In fact, this is the subject of current lines of research in the lab, but we do not feel that to precisely characterize these cellular events leading or concurrent to GPA,

and the possible roles of other meristem genes or hormonal markers (altogether, a whole new project) is essential for this study. Here, our goal has been to describe a novel genetic pathway that affects GPA timing, and we believe that we provide enough evidence to prove that FUL, AP2 and WUS expression dynamics in the SAM are involved in GPA regulation. Still, we agree that including other markers of meristem function such as CLV3 and STM could add valuable information and reinforce our conclusions, so following the reviewer's suggestions, we have included a time course analysis of STM::GUS and CLV3::GUS reporters in wt and *ful* backgrounds (see sup. Fig 2b), which show that also STM and CLV3 levels are more persistent in *ful* than in wt SAMs, and that, interestingly, WUS decline in expression precedes that of STM or CLV3, suggesting that WUS downregulation sets the onset of GPA.

2. Continuing on my argument on the importance of taking a global view of tissue aging, instead the current study focuses too much on the timing of termination of one molecule that is WUSCHEL (WUS). First of all the images showing lack of WUS-GUS in (Fig. 1E) in wild type are not convincing because I do not see expression in floral meristems which are expected to express WUS as they are much younger than the SAM.

Response: Please see our previous response. Also, new pictures have been included (fig S2a) addressing WUS expression in ovules of growing carpels of already arrested plants. Please note that, in arrested meristems, floral meristems already formed at the moment of GPA do not continue growing (see Fig 1e), so it makes sense that WUS is no longer expressed in these arrested floral buds.

Yes even if we assume that WUS downregulation occurs, that alone can not explain a universal phenomenon such as GPA. In fact *wus* mutants do produce several shoots that do not senesce for several months.

Response: Thanks for bringing out this point. GPA and senescence are not equivalent processes, GPA preceding plant senescence by several days. In *wus* mutants, plants grow to a certain extent, but they do not form functional SAMs, and when related structures are produced, they terminate very early. We believe that the fact that *wus* mutants do not senesce readily is more related to vegetative growth and not so much to GPA. We also agree that WUS downregulation is probably only one of the events leading to GPA, and we are working to gain new insights in other factors involved, but, again, we expect this work will be the subject of subsequent studies.

Therefore, if FUL is acting through repression of AP2, analyzing AP2-mediated transcriptome may provide a global view of GPA. Is there anything known about AP2-regulated transcriptome or even the FUL-regulated transcriptome that may better explain involvement of FUL in GPA?

Response: Thanks for the suggestion, we agree it is very interesting. Actually, we have embarked in a study that aims to determine the transcriptomic response of AP2 induction directing GPA reversion (see fig 3c). From this study, we have identified additional putative players in GPA control, but these are still being validated and will be the subject of a following manuscript.

3. Molecular analysis showing that AP2 is directly repressed by FUL is interesting and important for understanding of the onset of GPA. However, I find that this analysis is incomplete as it is only based on ChIP expts. Authors must consider experiments that show direct binding (EMSA) and transient transfection assays to prove the point that AP2 is directly repressed by FUL.

Response: Following reviewer's suggestions, we have included both EMSA and transient transfection analyses that fully support our previous conclusions.

Reviewer #3 (Remarks to the Author):

Balanza et al. described a series of genetic analysis of events leading to the global proliferative arrest, flowering and developing seeds. It is an open question how signals from developing seeds and age-dependent factors are integrated during the process. They at first observed the *ful* mutant and its delayed GPA. Then they screened an EMS enhancer-suppressor mutagenesis on *ful-1* and isolated a line of which nucleotide substitution was mapped in the last exon of the AP2 gene. The line gave the clue to link FUL-AP2-WUS and fill the gap between the GPA, flowering and setting seeds. The paper has an impact to shed light on FUL role in the flower/seed formation, thus we would like to know further information.

Response: We thank the reviewer for finding our study interesting and stimulating and welcome the comment and suggestions made.

Major points:

The authors proposed a new model depicted in Fig.4. To reinforce the model, I dare to ask regarding to Figure 1. What happens if WUS::GUS patterns are checked in *ap2-170*, *ful-1/ap2-170* background?

Response: WUS::GUS could not be tested in *ful-1* backgrounds because the *ful-1* allele has a T-DNA with a GUS reporter inserted in the FUL 5'UTR and then it is GUS positive. Therefore, we have produced a time course of WUS expression in inflorescences of *wt*, *ful-1*, *ap2-170* and *ful-1 ap2-170* by whole mount RNA in situ hybridization (See fig 2). These analyses show how WUS expression is maintained for longer in all these backgrounds, as it could be expected according to our hypothesis.

I wondered if total seeds set in each line might be roughly constant, if products between number of mature fruits and seeds/fruit are calculated in Fig.1B. Any comment on this point?

Response: The effect of seed set in GPA has been related to a threshold, not a quantitative effect in previous reports (see Hensel et al , 1994). In figure b, it can be seen how *fer/+* mutants and the *wt* have approx. the same number of fruits, while *wt* has twice as many seeds than *fer/+*; *wt* and *hec3* have also similar number of fruits, while the number of seeds in *hec3* is around 2/3 of *wt*. For *ful* alleles, the total number of seeds produced by the *ful-1* mutant is actually higher than in *wt* (around 20% more) while in *ful-2* it is roughly the same. These data are in accordance to the threshold model and to the specific role of FUL in regulating GPA timing and not total seed

production. Moreover, the higher number of nodes produced by *ful* mutants when seed set is avoided (see fig 1d), supports an independent effect of *ful* mutations and fertility in GPA.

If the targeting of AP2 mRNA by miR172 is important, the authors could address the issue whether miR172-resistant AP2 gene could mirror the *ap2-170* phenotype.

Response: We agree with the reviewer that it is possible to prove this in this way, and we have already provided some data in the ms. that point in this direction, like for example the phenotype of 35S::AP2170 that mirrors that of 35S::AP2m3. We have not generated a pAP2::AP2m3 transgenic line in the *ap2* null background for this work, and at this point it would involve a considerable amount of time. However, we have a comparable line to that reported in figure 3c (a 35S::LhGR>>AP2m3 line) that upon induction reverts GPA very similarly to the one described in this study. This line has been used to investigate the transcriptomic response of arrested inflorescence apices to GPA reversion, and it will be described in a following manuscript (see our last response to point 2 raised by Reviewer#2). We prefer not to include these data in this study for simplicity, but if necessary, we could provide at least a picture of reactivated 35S::LhGR4>>AP2m3 plants.

Minor points:

P.3, last two sentences could be modified to be reader-friendly
> ... when transformed with a 35S::AP2170 construct (Fig.2D), but not when transformed with 35S::AP2 (Fig S5),

Response: Thanks for the suggestion. Some rephrasing has been done accordingly.

Reviewer #4 (Remarks to the Author):

In this manuscript Balanza and colleagues describe a regulatory circuit that involves FRUITFULL (FUL) and APETALA2 (AP2) as regulator of global proliferative arrest (GPA) in *Arabidopsis thaliana*. AP2 has previously been implicated in meristem maintenance (i.e. Wuerschum et al., 2006) but the current study goes beyond this previous knowledge and provides information on the molecular regulation of AP2 by the MADS-domain transcription factor FUL, specifically during GPA. Given the GPA can affect yield in crops the findings should be of interest to a wider readership even though the current manuscript does not provide any evidence that the mechanism identified in *Arabidopsis* is conserved in other species.

Response: We thank the reviewer for finding our study interesting and novel. It is true that we do not address functional conservation in other species, but we actually have some data that provide evidence of functional conservation of FUL in this role in other species (pea), that hopefully will be the subject of a following manuscript, now in preparation.

According to the model provided by the authors AP2 integrates signals from FUL and miR172. One prediction from this model is that reduced miR172 levels should result in elevated AP2 and indirectly delay GPA. miR172 is encoded by several genes in

Arabidopsis thaliana, so a detailed genetic analysis is not really possible. However, I think that knocking down miR172s by a mimicry approach could be very informative. The prediction would be that reduced miR172 (possibly by expressing MIM172 from the FUL promoter) in a *ful* mutant should further delay GPA.

Response: We thank the reviewer for this suggestion. Actually, we have produced a number of transgenic lines expressing MIM172 and other miR172 resistant versions of AP2-like genes in the SAM that fully support our model (and the reviewer's prediction). While FUL promoter is not useful for this purpose (it also drives expression to the floral meristem and therefore it would likely cause floral indeterminacy), we have included in our approach other SAM promoters, such as TFL1, and, in most cases, GPA is delayed in these lines. However, we strongly prefer not to include these experiments in this manuscript, as they are the main body of a following study addressing the biotechnological applications (increased yield) that can be derived from our work. We believe that the evidence provided in the current manuscript is enough to identify FUL and AP2 as major players in GPA control and we prefer to keep the focus of the paper there.

The manuscript is well written but very concise. More details and background information should be provided, if space permits. Also, the figures not always fully support the claims made in text (see specific comments and questions provided below for details).

Response: We have tried to keep a concise style for simplicity, but we have taken in consideration the reviewer's following suggestions and are thankful for them.

Specific comments and questions:

Introduction

- page 2, line 2: introduce the abbreviation GPA also at first use in the main text, not only in the abstract
- page 2, line 4: sentence seems to be incomplete. Maybe "... optimize allocation of resources..."

R:Done

Results

- "ful mutants have delayed GPA", page 2:

The authors claim that the reduced seed set observed in *ful* mutants is not the cause of the GPA because two other mutants with reduced seed set, *fer* and *hec3*, do not show GPA. However, the number of different mutants examined (3) is rather small and I wonder how specific the GPA phenotype in *ful* really is.

Response: We have characterized two independent mutants for this study in different backgrounds and with seed set % below that of *ful* but above the proposed threshold of around 30% of wt (Hensel et al, 1994, see table V in this paper for other examples of partially sterile mutants characterized for GPA). We have shown that *fer/+* and *hec-3* mutants, unlike *ful*, behave as wt in terms of GPA. It is true that only two cases were included, but together with the reported examples by Hensel et al, and the experiment in fig 1d, the evidence clearly indicates that *ful* mutants have delayed GPA independently of seed production. Are there other mutants with altered GPA behavior? Certainly, but we still have to identify them and relate them to this process, which we hope will be one

of our goals in the near future.

- “WUS repression in the SAM correlates with GPA”, page 3:

GUS reporter at best give semi-quantitative information about gene expression and while the data provided on WUS expression being maintained in *ful* mutants is most likely correct I would feel more comfortable if the authors could provide independent evidence for this. i.e. RNA in situ hybridization (which I realize are also semi-quantitative at best) or qRT-PCR (which could be tricky as WUS is also expressed in floral primordia)

Response: Thanks for the suggestion. We have included WUS RNA in situ hybridization in the revised manuscript, that also support our previous hypothesis and extends these conclusions to the *ap2-170* and *ful-1 ap2-170* backgrounds.

- “FUL regulates GPA by fine tuning AP2 activity in the SAM”

page 3, line 4: “... screened an EMS enhancer-suppressor mutagenesis on *ful-1*.” Should be changed to “... performed an EMS enhancer-suppressor mutagenesis screen on *ful-1*.”

R:Done

page 3, line 6: the subject seems to be missing in the sentence “In addition, showed a strong...”. Possible “In addition, the mutant showed a strong...”

R:Done

page 3, line 12: mutations in the miR172 binding site of AP2 (-like) transcription factors (at least when expressed at high levels) have been shown to delay flowering. Does the *ap2-170* mutant flower later than wildtype? If it does, how might this be related to delayed GPA in *ful*?

Response: Under our conditions, *ap2-170* mutants flower very similarly to wildtype, so it is unlikely that this has any effect in GPA

page 3, line 13, 16: No conclusion on homeotic phenotypes in the outer three whorls possible based on the pictures provided in Fig 2B as this panel only shows fruits, not flowers. Please provide pictures of expanded flowers to support the claim that *ap2-170* protein is largely functional.

R: The relevant pictures have been included in Fig3b

page 3, line 21: The author write that “AP2 induction in senescent wildtype plants that already had undergone GPA restored the activity of the SAM...” How did the authors ensure that the SAM had already undergone GPA?

Response: As it can be seen in fig3c, the inflorescence shows evident signs of GPA (dehisced dry fruits contiguous to arrested floral buds). We just relied on this morphological evidence.

page 3, line 27: How did the authors confirm over-expression of AP2 in these lines? In particular in the 35S::AP2 line which apparently did not show any phenotypes.

Response: We did not perform a careful analysis (we expected the overexpressed wildtype form of AP2 would be degraded by miR172 and therefore assumed that expression analyses would not be too informative), but more than 30 T1 lines were generated, none with indeterminacy phenotypes, although around 15% of them showed

co-suppression phenotypes. In the case of 35S::AP2-170 lines, around 60% showed indeterminacy phenotypes, and 25% co-suppression phenotypes. Since the constructs were generated identically (same vector, only different by the reported mismatch in the AP2 CDS), we assumed their similar behavior. We hope this reasoning is convincing enough to support our claims.

page 4, line 4: The authors should provide expression data for other meristem-expressed genes, to show that the elevated expression was specific to AP2 and not caused by general differences in meristem size/architecture (which has been described for other Pap2 alleles; e.g. Wuerschum et al, 2006).

Response: While we agree with the reviewer that differences in SAM structure are a possibility, we have not seen evident differences in SAM size of wt and *ful* mutants (see figure S1), so this scenario is unlikely. Also, we are not claiming that AP2 is the only gene with higher expression in *ful* mutants (see for example fig S2b), just that the role of FUL in GPA regulation is mediated by AP2 (and AP2-like genes), claim that is backed up by genetic data (see Fig 4). In addition, it would be difficult to be systematic in choosing other candidate genes for this type of analyses, and moreover, dissecting SAMs for qRT-PCR is not trivial. For all these reason, we prefer not to proceed with these experiments at this point and we hoped to have convinced the reviewer on this not being necessary.

page 4, line 25: Fig 2F (?)

R: not in the revised ms.

page 4, line 30: "...observe frequent formation of three-carpel gynoecia...". Can this phenotype be quantified?

Response: A sentence was added to Fig S7 legend.

page 4, line 33: Fig S7 is fragmented, uses different fonts, and is generally not very nice. Please reassemble figure

Response: Thanks. We have produced a new version of the figure that certainly looks nicer.

page 5, line 2: "...so we analysed our ChIP-seq experiment in FUL::FUL:GFP inflorescences...". Which ChIP-seq experiment is this? Please provide reference or experimental details.

Response: In the revised ms. we have included a new ChIP-seq experiment. The text has been rewritten, both in the main body of the ms. and in the supplementary material, and we think now it is clearer.

page 5, line 6: "...while no significant FUL binding was observed in any of the MIR172 genes...". Even though this negative evidence the actual ChIP-seq data on miR172 not being bound by FUL should be included in Fig S8 to substantiate the claims made. The same font should be use for figures throughout the manuscript (incl. supplementary material).

R: Thanks. We have now included these data in Fig S5a and included new data in Fig 4.

- "AP2 and other miR172-regulated AP2-like genes participate in GPA regulation"

page 5, line 7: "...GPA occurred significantly earlier in the ap2 ...". No significance level indicated in Fig 3B. Which statistical test was used?

R: We used t-test, now included in the figure and figure legend.

page 5, line 16: Font color

R: We used color to indicate significant differences in seed set, but in the revised manuscript we have changed it to shadowing.

Figure legends:

Fig 1A: “ful mutants produce more fruits than WT”. This is actually not clearly visible from the figure. What can be seen is that the ful inflorescence is taller but the number of fruits is not clear.

R: True, but together with figure 1b, the claim is supported. Panel a is probably not necessary, but is quite graphic. We prefer to leave it like this for clarity.

Fig 2B: “ap2-170/ap2-2 plants do not show the typical homeotic defects of ap2-2 mutants”. The pictures show only fruits but not whorls 1 and 2 which are normally strongly affected in ap2 mutants. Please provide pictures of whole open flowers (or avoid talking about “typical homeotic defects”).

R: We have modified the figure and the legend according to reviewer’s suggestions.

Reviewers' comments:

Reviewer #1 (Remarks to the Author):

First of all I would like to thank to the authors for they efforts in addressing all the concerns from the reviewers, including myself. I would like to ask them to calculate the difference in total seed production between the two ful alleles and the corresponding WT backgrounds and the other mutant lines used (fer and hec3). By the numbers in Fig 1B, it seems very similar to me that the number of seeds produced is roughly similar since ful mutants produce more fruits but less seeds. If FUL is really involved in GPA and GPA has been related to fruit and seed production, they should find significant differences to conclude its direct role in GPA control. On the contrary, if those differences are not as significant as found in the other mutants shown within that figure, FUL is regulating, as shown before, fruit development which co-laterally might give the impression it is regulating GPA. It is possible that the higher amount of fruits it is a compensatory event due to the lower amount of seeds/per fruit rather that a GPA related phenomena.

That point is also important to interpret the results from the expression profile of the meristem marker genes presented in Fig 2 and Fig 2S. That their expression is longer maintained might be just a by side effect derived from a smaller number of seeds produced in ful mutants. Additionally, both in situ and gus staining are not quantitative techniques. Therefore, the authors should address that point and provide a quantitative measure over time.

Regarding the ChIP-seq experiments, The authors should provide that in those samples, FUL binding to the AP2 and AP2-like genes has any effect on their expression in addition to the results shown in Fig 4E from ful mutants compared to WT. That is especially important since in the luciferase assays, the presence of AP2 does not lead to significant changes unless they are fused to VP16 and despite of AP2 being under the strong 35S promoter. Transactivation assays in Nicotiana, should be done using the entire AP2 promoter and, ideally, compared with mutations in, at least, the two cis elements found to be enriched in the ChIP-seq assays. EMSAs should be performed with mutated probes or competition with cold probes to show specificity.

Reviewer #2 (Remarks to the Author):

Authors have addressed most of my earlier concerns. They have tested few other markers -STM and CLV3 which largely follow WUS pattern showed earlier. Authors have added new EMSA and transient analysis data to show that AP2 is directly repressed by WUS which strengthen the manuscript.

Reviewer #3 (Remarks to the Author):

I was satisfied to see the revised ms since the data (in situ hybridization, ChIP-qPCR) were placed and organized properly following the logical flow.

I noticed errors in the Figure.

Figure 4d: I could see only c and F at the bottom of lanes, not e or F as described in the Legend.

Figure 4f: I could see a, b, c at the top of ful-2 lane and a at the top of ful ap2 lane. Possibly they should appear in the reciprocal positions considering the statistical significance meaning.

Reviewer #4 (Remarks to the Author):

In the revised manuscript, Balanza and colleagues have addressed many of the comments and concerns I had concerning the original manuscript. In particular, the authors have added a whole mount RNA in situ experiment, which shows maintained expression WUS expression in the ful mutant.

As stated before, the manuscript, even though well written, is very compact, and I still feel that it would benefit the non-expert reader if more details and background information were provided. Also, to me it sometimes seems as if the authors overstate the facts (see comments below).

However, the authors should be able to address these concerns easily enough without any elaborate experiments by rephrasing the text (or providing additional data to support the claims made).

Specific comments:

Title page:

- Name of (at least) one of the authors misspelled: Kerstin Kaufmann

Abstract:

- Consider rephrasing "...WUSCHEL expression, an essential factor for stem cell production, ..." While not incorrect as such, WUS is more often referred to as a factor required for SAM maintenance, rather than being required for stem cell production.

Main text:

- Page 2, line 10: "Little is known about the molecular bases of GPA timing. It is well established that..."

I would suggest to change the order: first mention that fruits and seeds are important for meristem arrest, than mention that this is genetically controlled. Finally end with the statement that little is known about the molecular/genetic mechanisms controlling GPA and that the current manuscript sheds some light on this aspect.

- Page 2, line 31/32: I still wonder if *hec3* and *fer/+* mutants are the best control...

- Page 3, line 1: the authors write that "...number of flowers in ful mutants is specifically related to 1 FUL loss-of-function..."

On the other hand, the ful mutant still responds to removal of flowers by increased/maintained flower initiation (similar number than WT). Doesn't this indicate that there are factors other than FUL that control GPA? Could this be AP2, which also seems to act partially redundant to FUL (as indicated by the double mutant phenotype)?

- Page 3, line 9/10: here the authors present their new results from whole mount in situ hybridization.

While the results are most likely correct, I would recommend that the authors add a short "disclaimer", stating that in situ are notoriously difficult to quantify but that quantification of WUS expression in the SAM was not possible because of its expression in flower primordia and fruits/ovules, which makes i.e. qRT-PCR next to impossible.

- Page 3, line 32/33: "...but no other defects related to the described roles of AP2 in floral organ development or floral determinacy..."

ap2-170 fruits appear shorter than wt. is this a real phenotype and if yes, has this been quantified? What is the seed number in *ap2-170*. How does this compare to ful alleles or *hec3* or *fer/+*?

- Page 4, line 8/9: "...the dramatic delay in GPA and strong floral indeterminacy phenotype was only observed in ful background..."

According to Fig 3a fruit number is also significantly increased in the ap2-170 single mutant? So why is it "...only observed in the ful...". Please clarify

- Page 4, line 13/14: "...35S::AP2 did not cause visible defects in any of these backgrounds (Fig. 3d)..."

Has the expression of the AP2 transgene been confirmed? Just to make sure this is not caused by transgene silencing.

- Page 4, line 18: "... following removal of all visible floral buds under the scope."

Here it is difficult to judge, what the authors consider "visible". ;Stage 8 flowering, stage 6 flowers, or smaller? Please provide more accurate information here as the composition of the tissue might affect qRT-PCR results. Best would be SEM pictures of WT and ful inflorescences, indicating which tissues were used.

- Page 4, line 28/29: "No enrichment could be observed at the WUS locus (Fig. S5), indicating that FUL regulates WUS via AP2."

I do not agree with this statement. All the data really show is that WUS is not a direct FUL target. The data do not support the claim that FUL regulates WUS via AP2. For this the authors would need to show that regulation of WUS by FUL is compromised in an ap2 mutant. To substantiate their claims the authors could check available data sets on targets of AP2 (-like) proteins. If WUS is among these targets this would substantiate their claim that FUL regulates WUS via AP2

- Page 5/ line 1: "GUS activity was detected 1 at higher levels in ful..."

To me it seems as if GUS is more broadly expressed rather than that it is detected at higher levels. Difficult to say, really, as GUS is semi-quantitative at best..

Also it doesn't help that the panels present different magnifications (at least that is my impression), which makes comparisons difficult.

- Page 5/ line: "...elevated levels of AP2 in ful meristems could still be explained by two types of interactions:"

Not just interactions. To me it seems at least theoretically possible that the meristems are different between WT and ful and that there are simply more cells expression AP2 in the sampled tissue, hence the apparently higher "expression". Can the authors exclude this possibility?

- Page 5, line 14/15: Images provided are really clear. See previous comment on GUS activity.

- Page 5, line 15: Fig. 4b?

- Page 6, line 8: The authors should discuss why SNZ is not bound in vitro EMSA but bound and regulated in planta. How do the authors explain this discrepancy?

- Page 6, line 16: "...indicating that AP2 loss-of-function caused early SAM termination..."

I find this sentence little misleading as the data clearly show that ap2-12 produces as many fruits as wt, not less. It is only early compared to the nga quad. Please phrase more conservatively.

Discussion:

- Page 6, line 31: "...which in turn likely control WUS temporal maintenance in the SAM."

I strongly recommend that the authors check available AP2 ChIP data if WUS is a direct AP2 target. If

they do not find evidence of this, this would not invalidate the main conclusion, but this finding should be discussed more carefully, as effects of AP2 on WUS might be rather indirect. If this is the case, Fig 5 should be changed to reflect this.

Reviewer #1 (Remarks to the Author):

First of all I would like to thank to the authors for they efforts in addressing all the concerns from the reviewers, including myself.

Response: We thank the reviewer for appreciating our effort to address all the issues previously raised. We believe that now our study is stronger and more convincing.

I would like to ask them to calculate the difference in total seed production between the two *ful* alleles and the corresponding WT backgrounds and the other mutant lines used (*fer* and *hec3*). By the numbers in Fig 1B, it seems very similar to me that the number of seeds produced is roughly similar since *ful* mutants produce more fruits but less seeds. If *FUL* is really involved in GPA and GPA has been related to fruit and seed production, they should find significant differences to conclude its direct role in GPA control. On the contrary, if those differences are not as significant as found in the other mutants shown within that figure, *FUL* is regulating, as shown before, fruit development which co-laterally might give the impression it is regulating GPA. It is possible that the higher amount of fruits it is a compensatory event due to the lower amount of seeds/per fruit rather that a GPA related phenomena.

Response: This concern was previously raised for the first version of the ms (rev#3). We believe that our previous response to Reviewer 3 is equally valid here. Please note that, actually, it served to convince R#3. Below, we paste out previous explanation:

"The effect of seed set in GPA has been related to a threshold, not a quantitative effect in previous reports (see Hensel et al, 1994). In figure 1b, it can be seen how *fer/+* mutants and the wt have approx. the same number of fruits, while wt has twice as many seeds than *fer/+*; wt and *hec3* have also similar number of fruits, while the number of seeds in *hec3* is around 2/3 of wt. For *ful* alleles, the total number of seeds produced by the *ful-1* mutant is actually higher than in wt (around 20% more) while in *ful-2* it is roughly the same. These data are in accordance to the threshold model and to the specific role of *FUL* in regulating GPA timing and not total seed production. Moreover, the higher number of nodes produced by *ful* mutants when seed set is avoided (see fig 1d), supports independent effects of *ful* mutations and fertility in GPA."

That point is also important to interpret the results from the expression profile of the meristem marker genes presented in Fig 2 and Fig 2S. That their expression is longer maintained might be just a by side effect derived from a smaller number of seeds produced in *ful* mutants.

Response: As noted above, the effect of seed set in GPA has already been shown to be related to a threshold, not controlled by a quantitative effect (Hensel et al, 1994). It is therefore highly unlikely that the modest reduction in seed production per fruit in *ful* mutants has this effect.

Additionally, both *in situ* and *gus* staining are not quantitative techniques. Therefore, the authors should address that point and provide a quantitative measure over time.

Response: It is certainly true that these techniques are only semi-quantitative, but the *WUS* expression time courses, with two independent techniques (*in situ* and *GUS*), performed carefully with the same experimental conditions and in parallel for the different genetic backgrounds, show very similar and repetitive results, where differences in the strength of the signal are evident. We believe that these results are more convincing and reflect better meaningful differences in *WUS* expression than qRT-PCRs. For qRT-PCR analyses in SAMs, dissected tissue would have to be used (to avoid incorporating as much as possible the expression in floral buds) and this is already technically challenging in proliferating SAMs (we have actually done this to quantify *AP2* expression in wt/*ful*, see fig 3e, fig 4e), but virtually impossible in SAMs close to GPA (see Fig S1 to note the minuscule SAM size in older inflorescences), so we don't think that these experiments would help to improve our analyses. Please note that reviewer #4, noting that *in situ* are only semi-quantitative but qRT-PCRs are not a viable option, just recommends modifying the text accordingly, which we have done in the current revised version (see R#4 comment on Page 3, line 9/10)

Regarding the ChIP-seq experiments, the authors should provide that in those samples, FUL binding to the AP2 and AP2-like genes has any effect on their expression in addition to the results shown in Fig 4E from *ful* mutants compared to WT.

Response: Please note that, in this new version, the ChIP experiments have been done in *ap1 cal* background, where proliferative meristems are inflorescence meristems, and therefore floral tissues are not present (this way we assess FUL binding in the relevant tissues). To check the effect of *ful* mutation on the expression of AP2-like genes in this background, we should compare *ap1 cal* to *ap1 cal ful* mutants. However, in the triple *ap1 cal ful* mutants, the proliferative meristems have vegetative characters rather than inflorescence identity (please see Ferrandiz et al., 2000, *Development* 127: 725-34 for detailed characterization of the *ap1 cal ful* background), so the comparison would not be biologically equivalent. We believe that quantification of AP2, AP2-like genes and MIR172 genes in dissected inflorescence apices of wt and *ful* plants is much more adequate to evaluate the effect of FUL regulation of these genes in the SAM and that comparing *ap1 cal* to *ap1 cal ful* apices would not provide meaningful data.

That is especially important since in the luciferase assays, the presence of AP2 does not lead to significant changes unless they are fused to VP16 and despite of AP2 being under the strong 35S promoter. Transactivation assays in *Nicotiana*, should be done using the entire AP2 promoter and, ideally, compared with mutations in, at least, the two cis elements found to be enriched in the ChIP-seq assays. EMSAs should be performed with mutated probes or competition with cold probes to show specificity.

Response: The fact that FUL does not have a significant effect on AP2 promoter unless fused to VP16 is not surprising. FUL does not have activation domains or repression domains (at least none has been described and characterized), and, like many MADS factors, very likely relies of interaction with other proteins to regulate its targets. When fused to VP16 it is expected that activates transcription of genes bound directly by FUL independently of interactions with these other factors. We strongly believe that using the full AP2 promoter would not add any new information. First, the full AP2 promoter has not been characterized, we don't know exactly its size and functional elements; second, using fragments where we detected FUL binding and fragments where we didn't, and observing the activity of FUL only in the former cases (fragments bound) strongly supports that FUL bind to these fragments *in vivo* specifically; this also applies to argue that we do not think that mutating cis elements in these fragments is important: already in the non-bound fragment (AP-III in figure 4c) we can see that FUL:VP16 has no effect, again supporting specificity of binding. This relates also to the EMSA analyses. For the EMSAs, we used a master mix containing the FUL homodimers, and added this to the different probes. FUL is clearly binding only to a few probes, while the results for the other fragments are negative. It would be a lot of additional work to generate for every probe also a mutated probe, and our experience with previous EMSAs is that deletion of the CArG box always results in a loss of binding (e.g., see Bemer et al., *JXB* 2017). Deletion of the CArG box is useful if we would like to prove that FUL is really binding to a particular CArG box, but we just want to show here that FUL can bind to the regulatory regions of AP2-like genes, and the current experiment is sufficient for that.

In summary, we believe that, taken together, the new ChIP analyses (highly enriched in inflorescence meristem tissue), the expression analyses, the Luciferase assays and the EMSAs provide enough evidence to support FUL direct regulation of AP2 and A22-like genes and we hope to have convinced reviewer #1 on this.

Reviewer #2 (Remarks to the Author):

Authors have addressed most of my earlier concerns. They have tested few other markers-STM and CLV3 which largely follow WUS pattern showed earlier. Authors have added new EMSA and transient analysis data to show that AP2 is directly repressed by WUS which strengthen the manuscript.

Response: We thank the reviewer for his/her nice previous suggestions and for finding that the current version merits publishing.

Reviewer #3 (Remarks to the Author):

I was satisfied to see the revised ms since the data (in situ hybridization, ChIP-qPCR) were placed and organized properly following the logical flow.

I noticed errors in the Figure.

Figure 4d: I could see only c and F at the bottom of lanes, not e or F as described in the Legend.

Figure 4f: I could see a, b, c at the top of ful-2 lane and a at the top of ful ap2 lane. Possibly they should appear in the reciprocal positions considering the statistical significance meaning.

Response: We thank the reviewer for his/her nice previous suggestions and for finding that the current version has improved to merit publishing. Also, we are grateful for pointing out the errors in Figure 4, which have been now corrected.

Reviewer #4 (Remarks to the Author):

In the revised manuscript, Balanza and colleagues have addressed many of the comments and concerns I had concerning the original manuscript. In particular, the authors have added a whole mount RNA in situ experiment, which shows maintained expression WUS expression in the ful mutant.

As stated before, the manuscript, even though well written, is very compact, and I still feel that it would benefit the non-expert reader if more details and background information were provided. Also, to me it sometimes seems as if the authors overstate the facts (see comments below).

However, the authors should be able to address these concerns easily enough without any elaborate experiments by rephrasing the text (or providing additional data to support the claims made).

Response: We would like to thank the reviewer for the careful revision of the text and the suggestions to improve it. We are pleased to see that he/she is mostly satisfied with the new evidence introduced in this version, and that no new experiments are required. We have now taken into account the reviewer's comments to modify the text.

Specific comments:

Title page:

- Name of (at least) one of the authors misspelled: Kerstin Kaufmann
OK

Abstract:

- Consider rephrasing "...WUSCHEL expression, an essential factor for stem cell production, ..."
While not incorrect as such, WUS is more often referred to as a factor required for SAM maintenance, rather than being required for stem cell production.
OK

Main text:

- Page 2, line 10: "Little is known about the molecular bases of GPA timing. It is well established that..."
I would suggest to change the order: first mention that fruits and seeds are important for meristem arrest, then mention that this is genetically controlled. Finally end with the statement that little is known about the molecular/genetic mechanisms controlling GPA and that the current manuscript sheds some light on this aspect.
OK

- Page 2, line 31/32: I still wonder if hec3 and fer/+ mutants are the best control...

Response: We cannot assess whether they are the best controls, but in our opinion, they are good enough: The two mutations are unrelated, are in different genetic backgrounds, the percentage of seed-set reduction is also different (around 50% in *fer/+*, 35% in *hec3*, both less fertile than *ful*) and the basis of the poor fertility are completely unrelated (female gametophyte defect in *fer/+*, reduced transmitting tissues in *hec3*).

- Page 3, line 1: the authors write that "...number of flowers in *ful* mutants is specifically related to 1 FUL loss-of-function..."

On the other hand, the *ful* mutant still responds to removal of flowers by increased/maintained flower initiation (similar number than WT). Doesn't this indicate that there are factors other than FUL that control GPA? Could this be AP2, which also seems to act partially redundant to FUL (as indicated by the double mutant phenotype)?

Response. Our interpretation is that the seed-signal that promotes GPA is still working in *ful* mutants (seed removal delays both WT and *ful*), so FUL controls GPA independently of seed production (in the age pathway). Also, this result (together with the comparison with partially sterile mutants) argues against the possibility that total number of seeds in *ful* fruits are important for GPA control. It is important to note, also, that we are not claiming at any moment that FUL and AP2 are the only factors controlling GPA, we only say they are important for this regulation, and that FUL acts on AP2 and AP2-like genes (which would be then the redundant functions). It is very likely than new factors, both in the seed-dependent and the age-dependent pathway, will be identified soon, but they are obviously not the focus of this study.

- Page 3, line 9/10: here the authors present their new results from whole mount in situ hybridization.

While the results are most likely correct, I would recommend that the authors add a short "disclaimer", stating that in situ are notoriously difficult to quantify but that quantification of WUS expression in the SAM was not possible because of its expression in flower primordia and fruits/ovules, which makes i.e. qRT-PCR next to impossible.

Response: We have modified the text accordingly.

- Page 3, line 32/33: "...but no other defects related to the described roles of AP2 in floral organ development or floral determinacy..."

ap2-170 fruits appear shorter than wt. is this a real phenotype and if yes, has this been quantified? What is the seed number in *ap2-170*. How does this compare to *ful* alleles or *hec3* or *fer/+*?

Response: Certainly, *ap2-170* fruits are aprox. 20% shorter than wt (only that, to our knowledge, this phenotype was never described for *ap2* mutants before). Unfortunately, we do not have a careful quantification of *ap2-170* seed number per fruit in the same way as that presented in figs 1b or 4f, but we observed that the fruits did not show any sign of seed abortion or partial fertility, and seed set was comparable to wildtype, way above the proposed threshold required to affect GPA (please note that only plants with fruits below 30-35% of seed set show delayed GPA - Hensel et al, 1994, and our own data). We have modified the text to incorporate these observations and made this clear.

- Page 4, line 8/9: "...the dramatic delay in GPA and strong floral indeterminacy phenotype was only observed in *ful* background..."

According to Fig 3a fruit number is also significantly increased in the *ap2-170* single mutant? So why is it "...only observed in the *ful*...". Please clarify

Response: we considered that the delay in GPA was only dramatic in the *ful* background, although also significant in FUL wt. Since it is just a language problem, we have modified the text accordingly.

- Page 4, line 13/14: "...35S::AP2 did not cause visible defects in any of these backgrounds (Fig. 3d)..."

Has the expression of the AP2 transgene been confirmed? Just to make sure this is not caused by transgene silencing.

Response: we provided a detailed response to a related comment to the previous version, justifying why we judge this unnecessary given the number of independent lines tested for 35S::AP2 and 35S::AP2¹⁷⁰ transgenes. Still, we recently did a quick semiquantitative RT-PCR

test on seedlings (so miR172 levels are low) from five random 35S::AP2 T2 lines and they all show elevated AP2 expression compared to the WT control (not shown).

- Page 4, line 18: "... following removal of all visible floral buds under the scope." Here it is difficult to judge, what the authors consider "visible". ; Stage 8 flowering, stage 6 flowers, or smaller? Please provide more accurate information here as the composition of the tissue might affect qRT-PCR results. Best would be SEM pictures of WT and *ful* inflorescences, indicating which tissues were used.

Response: a more accurate description has been added to the methods section.

- Page 4, line 28/29: "No enrichment could be observed at the WUS locus (Fig. S5), indicating that FUL regulates WUS via AP2." I do not agree with this statement. All the data really show is that WUS is not a direct FUL target. The data do not support the claim that FUL regulates WUS via AP2. For this the authors would need to show that regulation of WUS by FUL is compromised in an *ap2* mutant. To substantiate their claims the authors could check available data sets on targets of AP2 (-like) proteins. If WUS is among these targets this would substantiate their claim that FUL regulates WUS via AP2

Response: Thanks for bringing this up. Certainly, these results only show that FUL regulates WUS indirectly. Actually, even though AP2 has been described to regulate WUS (refs), binding of AP2 or AP2-like genes to the WUS promoter has not been reported, so it is likely that AP2 also controls WUS expression indirectly. We think that we can address this issue and avoid overstatement by reformulating this sentence to say that FUL controls GPA timing, which is tightly correlated with WUS expression maintenance in the SAM, through AP2-like genes, a point that is strongly supported by our genetic analyses (GPA delay in *ful* is fully reverted in a 35S::miR172 background, see figure 4f) and molecular data. Actually, this is the main message of the paper, as reflected by the title: "Genetic control of meristem arrest and life span in Arabidopsis by a novel *FRUITFULL-APETALA2* pathway", and throughout the text.

- Page 5/ line 1: "GUS activity was detected 1 at higher levels in *ful*..." To me it seems as if GUS is more broadly expressed rather than that it is detected at higher levels. Difficult to say, really, as GUS is semi-quantitative at best... Also it doesn't help that the panels present different magnifications (at least that is my impression), which makes comparisons difficult.

Response: Magnification in the panels is the same (compare floral bud size). GUS has been detected in parallel and with the same experimental conditions for all samples and GUS signal appears stronger and broader in *ful* and FUL:VP16. We agree that this does not reflect exact quantitative differences in AP2 expression, but it shows that *ful* or FUL:VP16 affect AP2 promoter activity. Since this is just one of the experiments that we provide to show FUL activity in AP2 promoter, we do not believe that this needs further clarification or modifications of the text.

- Page 5/ line: "...elevated levels of AP2 in *ful* meristems could still be explained by two types of interactions:"

Not just interactions. To me it seems at least theoretically possible that the meristems are different between WT and *ful* and that there are simply more cells expression AP2 in the sampled tissue, hence the apparently higher "expression". Can the authors exclude this possibility?

Response: No, we cannot, but it seems highly unlikely. We have no evidence of wt and *ful* meristems being different (see fig S1 SEM pictures, or the similar rate of flower production by *ful* and wt plants in fig 1c).

- Page 5, line 14/15: Images provided are really clear. See previous comment on GUS activity.

See previous response

- Page 5, line 15: Fig. 4b?

- Page 6, line 8: The authors should discuss why SNZ is not bound in vitro EMSA but bound and regulated in planta. How do the authors explain this discrepancy?

Response: It is possible that FUL binding to the different promoter depends to some extent of other interacting factors. Indeed, as seen in the EMSA assay, in some cases FUL appears to bind preferentially as a homodimer (AP2-I) and in others as a tetramer (AP2-II, pTOE3). Binding to SNZ promoter is detected by CHIP, in its native biological context, what makes likely that required interactor are present. A sentence has been included in the text to better discuss these results.

- Page 6, line 16: "...indicating that AP2 loss-of-function caused early SAM termination..."
I find this sentence little misleading as the data clearly show that *ap2-12* produces as many fruits as wt, not less. It is only early compared to the *nga* quad. Please phrase more conservatively.

Response: We see the reviewer's point, but we do think that the sentence is accurate, since it says "flower production in the loss-of-function *ap2-12* mutants, was much lower than in other similarly sterile mutants such as *nga1 nga2 nga3 nga4 (nga^{quad})*, indicating that AP2 loss-of-function caused early SAM termination". Please also note that when flowers are removed in wt and *ap2-12* backgrounds, therefore rendering both genotypes sterile, *ap2* plants terminate significantly earlier than wt. Still, we have modified slightly the sentence in the text

Discussion:

- Page 6, line 31: "...which in turn likely control WUS temporal maintenance in the SAM."
I strongly recommend that the authors check available AP2 ChIP data if WUS is a direct AP2 target. If they do not find evidence of this, this would not invalidate the main conclusion, but this finding should be discussed more carefully, as effects of AP2 on WUS might be rather indirect. If this is the case, Fig 5 should be changed to reflect this.

Response: Nowhere in the manuscript we had the intention to imply that the activity of AP2 on WUS regulation is direct. Our point is that FUL directly regulates AP2 and AP2-like genes and this module controls GPA timing, which is correlated with WUS being no longer expressed in the SAM. AP2 has been proposed as a WUS regulator in several other publications (most importantly in Zhao et al, Plant J, 2007 or Würschum et al, Plant Cell, 2006, but also, more recently, Huang et al, New Phytol, 2017, where an AG-independent role of AP2 in WUS temporal maintenance in the floral meristem is shown) but no direct AP2 binding to WUS promoter has been reported and it is currently unclear how AP2 regulates WUS expression.

We think that, throughout the text (check, for example, title and wrapping up sentence), we have been careful to restrict our main conclusion to the role of FUL/AP2 in GPA, which involves the temporal control of WUS expression in the SAM. It is not our goal to prove that AP2 directly regulates WUS (actually, it seems unlikely). Just to make it more clear, we have modified the text in page 4, L5 or page 7, L3. Because in figure 5 no direct molecular interactions are implied (for example seed/flower signal repressing WUS clearly does not represent a factor directly binding to WUS), we feel that the model is correct as a representation of factors controlling GPA and the positive/negative effect of their activities.